# What Do We Maximize in Self-Supervised Learning And Why Does Generalization Emerge?

## Abstract

In this paper, we provide an information-theoretic (IT) understanding of self-supervised learning methods, their construction, and optimality. As a first step, we demonstrate how IT quantities can be obtained for deterministic networks as an alternative to the commonly used unrealistic stochastic networks assumption. Secondly, we demonstrate how different SSL models can be (re)discovered based on first principles and highlight the underlying assumptions of different SSL variants. Based on this understanding, we present new SSL methods that are superior to existing methods in terms of performance. Third, we derive a novel generalization bound based on our IT understanding of SSL methods, providing generalization guarantees for the downstream supervised learning task. As a result of this bound, along with our unified view of SSL, we can compare the different approaches and provide general guidelines to practitioners. Consequently, our derivation and insights contribute to a better understanding of SSL and transfer learning from a theoretical and practical perspective.

## 1 Introduction

Self-Supervised Learning methods (SSL) learn representations using a surrogate objective between inputs and self-defined signals. In SimCLR (Chen et al., 2020), for example, a contrastive loss is defined that makes representations for different versions of the same image similar, while making the representations for different images different. After optimizing the surrogate objective, the pre-trained model is used as a feature extractor for a downstream supervised task, such as image classification, object detection, instance segmentation and transfer learning (Caron et al., 2021; Chen et al., 2020; Misra & Maaten, 2020; Shwartz-Ziv et al., 2022). However, despite success in practice, only a few number of authors (Arora et al., 2019; Lee et al., 2021a) have sought to provide theoretical insights about the effectiveness of SSL.

In recent years, information theory methods have played a key role in several deep learning achievements, from practical applications in representation learning (Alemi et al., 2016), to theoretical investigations (Xu & Raginsky, 2017; Steinke & Zakynthinou, 2020; Shwartz-Ziv, 2022). Moreover, different deep learning problems have been successfully approached by developing and applying novel estimators and learning principles derived from information-theoretic quantities. Specifically, many works have attempted to analyze SSL from an information theory perspective. An example is the use of the renowned information maximization (InfoMax) principle (Linsker, 1988) in SSL (Bachman et al., 2019). However, looking at these works may be confusing. Numerous objective functions are presented without a rigorous justification, some contradicting each other, as well as many implicit assumptions (Kahana & Hoshen, 2022; Wang et al., 2022; Lee et al., 2021b) Moreover, these works rely on a crucial assumption: a stochastic DN mapping, which is rarely the case nowadays.

This paper presents a unified framework for SSL methods from an information theory perspective, which can be applied to deterministic DN training. We summarize our contributions into four points: (i) First, in order to study deterministic DNs from an information theory perspective, we shift stochasticity to the DN input, which is a much more faithful assumption for current training techniques. (ii) Second, based on this formulation, we analyze how current SSL methods that use deterministic networks optimize information-theoretic quantities. (iii) Third, we present new SSL

methods based on our analysis and empirically validate their superior performance. (iv) Fourth, we study how the optimization of information-theoretic quantities is related to the final performance in the downstream task using a new generalization bound.

## 2 BACKGROUND

**Continuous Piecewise Affine (CPA) Mappings.** A rich class of functions emerges from piecewise polynomials: spline operators. In short, given a partition $\Omega$ of a domain $\mathbb{R}^D$, a spline of order $k$ is a mapping defined by a polynomial of order $k$ on each region $\omega \in \Omega$ with continuity constraints on the entire domain for the derivatives of order $0,\ldots,k-1$. As we will focus on affine splines ($k = 1$), we define this case only for concreteness. An $K$-dimensional affine spline $f$ produces its output via

$$f(\boldsymbol{z}) = \sum_{\omega \in \Omega} (\boldsymbol{A}_\omega \boldsymbol{z} + \boldsymbol{b}_\omega) \mathbb{1}_{\{\boldsymbol{z} \in \omega\}},$$ (1)

with input $\boldsymbol{z} \in \mathbb{R}^D$ and $\boldsymbol{A}_\omega \in \mathbb{R}^{K \times D}, \boldsymbol{b}_\omega \in \mathbb{R}^K, \forall \omega \in \Omega$ the per-region *slope* and *offset* parameters respectively, with the key constraint that the entire mapping is continuous over the domain $f \in \mathcal{C}^0(\mathbb{R}^D)$. Spline operators and especially affine spline operators have been widely used in function approximation theory (Cheney & Light, 2009), optimal control (Egerstedt & Martin, 2009), statistics (Fantuzzi et al., 2002), and related fields.

**Deep Networks.** A deep network (DN) is a (non-linear) operator $f_\Theta$ with parameters $\Theta$ that map a *input* $\boldsymbol{x} \in \mathbb{R}^D$ to a *prediction* $\boldsymbol{y} \in \mathbb{R}^K$. The precise definitions of DNs operators can be found in Goodfellow et al. (2016). We will omit the $\Theta$ notation for clarity unless needed. The only assumption we require for our study is that the non-linearities present in the DN are CPA, as is the case with (leaky-) ReLU, absolute value, and max-pooling. In that case, the entire input-output mapping becomes a CPA spline with an implicit partition $\Omega$, the function of the weights and architecture of the network (Montufar et al., 2014; Balestriero & Baraniuk, 2018). For smooth nonlinearities, our results hold from a first-order Taylor approximation argument.

**Self-Supervised Learning**. Joint embedding methods learn the DN parameters $\Theta$ without supervision and input reconstruction. The difficulty of SSL is to produce a good representation for downstream tasks whose labels are not available during training —while avoiding a trivially simple solution where the model maps all inputs to constant output. Many methods have been proposed to solve this problem, see Balestriero & LeCun (2022) for a summary and connections between methods. *Contrastive methods* learn representations by contrasting positive and negative examples, e.g. Sim-CLR (Chen et al., 2020) and its InfoNCE criterion (Oord et al., 2018). Other recent work introduced *non-contrastive methods* that employ different regularization methods to prevent collapsing of the representation. Several papers used stop-gradients and extra predictors to avoid collapse (Chen & He, 2021; Grill et al., 2020) while Caron et al. (2020) uses an additional clustering step. As opposed to contrastive methods, noncontrastive methods do not explicitly rely on negative samples. Of particular interest to us is the *VICReg* method (Bardes et al., 2021) that considers two embedding batches $\boldsymbol{Z} = [f(\boldsymbol{x}_1), \ldots, f(\boldsymbol{x}_N)]$ and $\boldsymbol{Z}' = [f(\boldsymbol{x}'_1), \ldots, f(\boldsymbol{x}'_N)]$ each of size $(N \times K)$. Denoting by $\boldsymbol{C}$ the $(K \times K)$ covariance matrix obtained from $[\boldsymbol{Z}, \boldsymbol{Z}']$ we obtain the VICReg triplet loss

$$\mathcal{L} = \frac{1}{K} \sum_{k=1}^{K} \left( \alpha \max \left(0, \gamma - \sqrt{\boldsymbol{C}_{k,k} + \epsilon}\right) + \beta \sum_{k' \neq k} (\boldsymbol{C}_{k,k'})^2 \right) + \gamma \|\boldsymbol{Z} - \boldsymbol{Z}'\|_F^2 / N.$$

**Deep Networks and Information-Theory.** Recently, information-theoretic methods have played a key role in several remarkable deep learning achievements (Alemi et al., 2016; Xu & Raginsky, 2017; Steinke & Zakynthinou, 2020; Shwartz-Ziv & Tishby, 2017). Moreover, different deep learning problems have been successfully approached by developing and applying information-theoretic estimators and learning principles (Hjelm et al., 2018; Belghazi et al., 2018; Piran et al., 2020; Shwartz-Ziv et al., 2018). There is, however, a major problem when it comes to analyzing information-theoretic objectives in deterministic deep neural networks: the source of randomness. The mutual information between the input and the representation in such networks is infinite, resulting in ill-posed optimization problems or piecewise constant, making gradient-based optimization methods ineffective (Amjad & Geiger, 2019). To solve these problems, researchers have proposed several solutions. For SSL, stochastic deep networks with variational bounds could be used, where the output of the deterministic network is used as parameters of the conditional distribution (Lee et al.,

2021b; Shwartz-Ziv & Alemi, 2020). Dubois et al. (2021) suggested another option, which assumed that the randomness of data augmentation among the two views is the source of stochasticity in the network. Another line of works assume a random input, but not using any properties of the distribution of the newtork's output in order to analysis the networkr's objective, which rely on general lower bounds (Wang & Isola, 2020; Zimmermann et al., 2021). For supervised learning, Goldfeld et al. (2018) introduced an auxiliary (noisy) DN framework by injecting additive noise into the model and demonstrated that it is a good proxy for the original (deterministic) DN in terms of both performance and representation. Finally, Achille & Soatto (2018) found that minimizing a stochastic network with a regularizer is equivalent to minimizing cross-entropy over deterministic DNs with multiplicative noise. However, all of these methods assume that the noise comes from the model itself, which contradicts current training methods. In this work, we explicitly assume that the stochasticity comes from the data, which is a less restrictive assumption and does not require changing current algorithms.

## 3 INFORMATION THEORY FOR DETERMINISTIC DEEP NETWORKS

This section first sets up notation and assumption on the information-theoretic challenges in SSL (section 3.1) and on our assumptions regarding the data distribution (section 3.2) so that any training sample $x$ can be seen as coming from a single Gaussian distribution as in $x \sim \mathcal{N}(\mu_x, \Sigma_x)$. From this we obtain that the output of any deep network $f(x)$ corresponds to a mixture of truncated Gaussian (section 3.3). In particular, it can fall back to a single Gaussian under small noise ($\det(\Sigma) \to \epsilon$) assumptions. These results will enable information measures to be applied to deterministic DNs. We then recover known SSL methods (Bardes et al., 2021; Chen et al., 2020) by making different assumptions about the data distribution and estimating their information.

### 3.1 SSL AS AN INFORMATION-THEORETIC PROBLEM

To better grasp the difference between key SSL methods, we first formulate the general SSL goal from an information-theoretical perspective. We start with the *MultiView InfoMax* principle, i.e., maximizing the mutual information between the representations. Let $X$ and $X'$ be two different views and $Z$ and $Z'$ their corresponding representations. As shown in Federici et al. (2020), to maximize their information, we maximize $I(Z; X')$ and $I(Z'; X)$ using the lower bound

$$I(Z, X') = H(Z) - H(Z|X') \geq H(Z) + \mathbb{E}_{x'}[\log q(z|x')] \tag{2}$$

where $H(Z)$ is the entropy of $Z$. In supervised learning, where we need to maximize $I(Z; Y)$, the labels ($Y$) are fixed, the entropy term $H(Y)$ is constant, and you only need to optimize the log-loss $\mathbb{E}_{x'}[\log q(z|x)]$ (cross-entropy or square loss). However, it is well known that for Siamese networks there exists a degenerate solution, in which all outputs "collapse" into an undesired value (Chen et al., 2020). Looking at eq. (2) we can see that the entropies are not constant and can be optimized throughout the learning process. Therefore, only minimizing the log loss will cause it to collapse to the trivial solution of making the representations constant (where the entropy goes to zero). To regularize these entropies, that is, prevent collapse, different methods utilize different approaches to implicit regularizing information. To recover them in section 4, we must first introduce the results around the data distribution (section 3.2) and how a DN transforms that distribution (section 3.3).

### 3.2 DATA DISTRIBUTION HYPOTHESIS

Our first step is to assess how the output random variables of the network are represented, assuming a distribution on the data itself. Under the manifold hypothesis, any point can be seen as a Gaussian random variable with a low-rank covariance matrix in the direction of the manifold tangent space of the data (Fefferman et al., 2016). Therefore, we will consider throughout this study the conditioning of a latent representation with respect to the mean of the observation, i.e., $X|x^* \sim \mathcal{N}(x^*, \Sigma_{x^*})$ where the eigenvectors of $\Sigma_{x^*}$ are in the same linear subspace than the tangent space of the data manifold at $x^*$, which varies with the position of $x^*$ in space.

Hence a dataset is considered to be a collection of $\{\boldsymbol{x}_n^*, n = 1, \ldots, N\}$ and the full data distribution to be a sum of low-rank covariance Gaussian densities, as in

$$X \sim \sum_{n=1}^{N} \mathcal{N}(\boldsymbol{x}_n^*, \Sigma_{\boldsymbol{x}_n^*})^{1\{T=n\}}, T \sim \mathrm{Cat}(N), \tag{3}$$

with $T$ the uniform Categorical random variable. For simplicity, we consider that the effective support of $\mathcal{N}(\boldsymbol{x}_i^*, \Sigma_{\boldsymbol{x}_i^*})$ and $\mathcal{N}(\boldsymbol{x}_j^*, \Sigma_{\boldsymbol{x}_j^*})$ do not overlap, where the effective support is defined as $\{x \in \mathbb{R}^D : p(x) > \epsilon\}$ This keeps things general, as it is enough to cover the domain of the data manifold overall, without overlap between different Gaussians. Therefore, we have that.

$$p(\boldsymbol{x}) \approx \mathcal{N}\left(\boldsymbol{x}; \boldsymbol{x}_{n(\boldsymbol{x})}^*, \Sigma_{\boldsymbol{x}_{n(\boldsymbol{x})}^*}\right) / N, \tag{4}$$

where $\mathcal{N}(\boldsymbol{x}; ., .)$ is the Gaussian density at $\boldsymbol{x}$ and with $n(\boldsymbol{x}) = \arg\min_n (\boldsymbol{x} - \boldsymbol{x}_n^*)^T \Sigma_{\boldsymbol{x}_n^*} (\boldsymbol{x} - \boldsymbol{x}_n^*)$. This assumption, that a dataset is a mixture of Gaussians with non-overlapping support, will simplify our derivations below, and could be extended to the general case if needed.

### 3.3 Data Distribution After Deep Network Transformation

Consider an affine spline operator $f$ (Eq. 1) that goes from a space of dimension $D$ to a space of dimension $K$ with $K \geq D$. The span, that we denote as image, of this mapping is given by

$$Im(f) \triangleq \{f(\boldsymbol{x}) : \boldsymbol{x} \in \mathbb{R}^D\} = \bigcup_{\omega \in \Omega} \mathrm{Aff}(\omega; \boldsymbol{A}_\omega, \boldsymbol{b}_\omega) \tag{5}$$

with $\mathrm{Aff}(\omega; \boldsymbol{A}_\omega, \boldsymbol{b}_\omega) = \{\boldsymbol{A}_\omega \boldsymbol{x} + \boldsymbol{b}_\omega : \boldsymbol{x} \in \omega\}$ the affine transformation of region $\omega$ by the per-region parameters $\boldsymbol{A}_\omega, \boldsymbol{b}_\omega$, and with $\Omega$ the partition of the input space in which $\boldsymbol{x}$ lives in. The practical computation of the per-region affine mapping can be obtained by setting $\boldsymbol{A}_\omega$ to the Jacobian matrix of the network at the corresponding input $x$, and $b$ to be defined as $f(x) - \boldsymbol{A}_\omega x$.

Therefore, the DN mapping consists of affine transformations on each input space partition region $\omega \in \Omega$ based on the coordinate change induced by $\boldsymbol{A}_\omega$ and the shift induced by $\boldsymbol{b}_\omega$.

When the input space is equipped with a density distribution, this density is transformed by the mapping $f$. In general, finding the density of $f(X)$ is an intractable task. However, given our disjoint support assumption provided in section 3.2, we can arbitrarily increase the representation power of the density by increasing the number of prototypes $N$. In doing so, the support of each Gaussian is included with the region $\omega$ in which its means lie in, leading to the following result.

**Theorem 1.** *Given the setting of eq. (4) the unconditional DN output density denoted as $Z$ is approximately a mixture of the affinely transformed distributions $\boldsymbol{x}|\boldsymbol{x}_{n(\boldsymbol{x})}^*$ e.g. for the Gaussian case*

$$Z \sim \sum_{n=1}^{N} \mathcal{N}\left(\boldsymbol{A}_{\omega(\boldsymbol{x}_n^*)} \boldsymbol{x}_n^* + \boldsymbol{b}_{\omega(\boldsymbol{x}_n^*)}, \boldsymbol{A}_{\omega(\boldsymbol{x}_n^*)}^T \Sigma_{\boldsymbol{x}_n^*} \boldsymbol{A}_{\omega(\boldsymbol{x}_n^*)}\right)^{1\{T=n\}}, \tag{6}$$

*where $\omega(\boldsymbol{x}_n^*) = \omega \in \Omega \iff \boldsymbol{x}_n^* \in \omega$ is the partition region in which the prototype $\boldsymbol{x}_n^*$ lives in.*

*Proof.* The proof of of Theorem 1 is presented in Appendix A. $\square$

## 4 Information Optimization and Optimality

Next, we will show how SSL algorithms for deterministic networks can be derived. According to Section 3.1, we want to maximize $I(Z; X')$ and $I(Z'; X)$. Although this mutual information is intractable in general, we can obtain a tractable variational estimation using the expected loss. First, when our input noise is small, namely that the effective support of the Gaussian centered at $x$ is contained within the region $w$ of the DN's input space partition, we can reduce the conditional output density to a single Gaussian: $(Z'|X' = x_n) \sim \mathcal{N}(\mu(x_n), \Sigma(x_n))$, where $\mu(x_n) = \boldsymbol{A}_{\omega(\boldsymbol{x}_n)} x_n + \boldsymbol{b}_{\omega(\boldsymbol{x}_n)}$ and $\Sigma(x_n) = \boldsymbol{A}_{\omega(\boldsymbol{x}_n)}^T \Sigma_{\boldsymbol{x}_n} \boldsymbol{A}_{\omega(\boldsymbol{x}_n)}$. Second, In order to compute the expected loss, we need

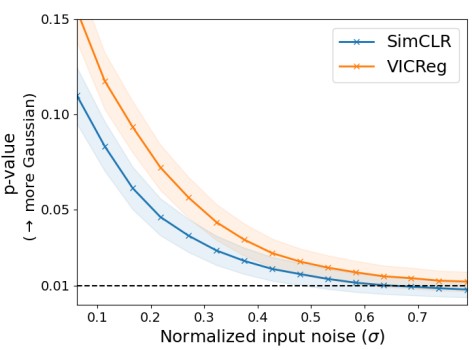
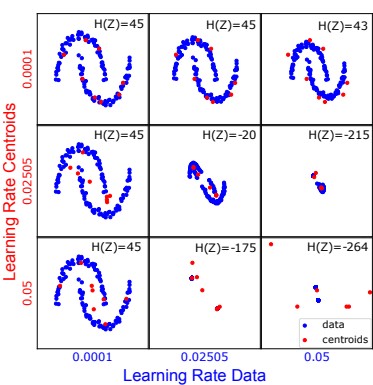

Figure 1: **Left: The network output SSL training is more Gaussian for small input noise**. The P-value of the normality test for different SSL models trained on CIFAR-10 for different input noise levels. The dashed line represents the point at which the null hypothesis (Gaussian distribution) can be rejected with $99\%$ confidence. **Right: Smaller learning rates prevent collapsing.** GMM points where in black is the entropy, in blue and red are the data points and GMM centroids respectively, with the corresponding learning rate

to marginalize out the stochasticity in the output of the network. In general, training with squared loss is equivalent to assuming a Gaussian observation model $p(z|z') \sim \mathcal{N}(z', \Sigma_r)$, where $\Sigma_r = I$. To compute the expected loss over samples of $x'$, we need to marginalize out the stochasticity in $Z'$: which means that the conditional decoder is a Gaussian - $(Z|X' = x_n) \sim \mathcal{N}(\mu(x_n), \Sigma_r + \Sigma(x_n))$. However, the expected log loss over samples of $Z$ is hard to compute, and therefore we focused on its lower bound, the expected log loss over samples of $Z'$. For simplicity, we set $\Sigma_r = I$ which gives us:

$$\mathbb{E}_{x'}\left[\log q(z|x')\right] \geq \mathbb{E}_{z'|x'}\left[\log q(z|z')\right] = \frac{d}{2}\log 2\pi - \frac{1}{2}\left(z - \mu(x')\right)^2 - \frac{1}{2}Tr\log\Sigma(x') \quad (7)$$

and now we can take the expectation over $Z$:

$$\mathbb{E}_{z|x}\left[\mathbb{E}_{z'|x'}\left[\log q(z|z')\right]\right] = \frac{d}{2}\log 2\pi - \frac{1}{2}\left(\mu(x) - \mu(x')\right)^2 - \frac{1}{2}\log\left(|\Sigma(x)| \cdot |\Sigma(x')|\right) \quad (8)$$

Full derivations of eq. (7) and eq. (8) are presented in Appendix B. Combine all the above give us

$$I(Z; X') \geq H(Z) + \mathbb{E}_{x,z|x,x',z'|x'}\left[\log q(z|z')\right] \quad (9)$$

$$= H(Z) + \frac{d}{2}\log 2\pi - \frac{1}{2}\mathbb{E}_{x,x'}\left[\left(\mu(x) - \mu(x')\right)^2 + \log\left(|\Sigma(x)| \cdot |\Sigma(x')|\right)\right] \quad (10)$$

To optimize it in practice, we can approximate $p(x, x')$ using the empirical data distribution:

$$L \approx \frac{1}{N}\sum_{i=1}^{N} H(Z) - \frac{1}{2}\left(\mu(x_i) - \mu(x_i')\right)^2 - \frac{1}{2}\log\left(|\Sigma(x_i)| \cdot |\Sigma(x_i')|\right) \quad (11)$$

Next, we will discuses how the estimation of the intractable entropy $H(Z)$ effect our objective.

### 4.1 DERIVING VICREG FROM FIRST PRINCIPLES

As a result of eq. (11), we can reconstruct VICReg from first principles. Unfortunately, $H(Z)$ cannot be determined explicitly. However, there are several approximations in the literature (Kolchinsky & Tracey, 2017; Huber et al., 2008). For a detailed discussion about the different entropy estimator, see appendix C. A simpler solution is to approximate the entire mixture by capturing the first two moments of the distribution, which provides an upper bound on the entropy. Note that we are optimizing an upper bound, which means we do not have a formal guarantee, and could lead to an arbitrary increase in our estimator. In practice, there are cases where we can achieve good results by maximizing a lower bound (Martinez et al., 2021; Nowozin et al., 2016), even though this may cause instability in the training process. Using $\Sigma_Z$ as the covariance matrix of $Z$, we will maximize:

$$L \approx \sum_{n=1}^{N} \log \frac{|\Sigma_Z|}{|\Sigma(x_i)| \cdot |\Sigma(x'_i)|} - \frac{1}{2}(\mu(x) - \mu(x'))^2 \qquad (12)$$

Optimizing the log determinate of $Z$ means maximizing its log eigenvalues. Although it is theoretically possible to differentiate eigendecomposition, this leads to numerical instability (Dang et al., 2018). While many works have attempted to address this issue (Giles, 2008; Ionescu et al., 2015), VICReg is using a straightforward approach. Because the eigenvalues of a diagonal matrix are the diagonal, increasing the sum of the log-diagonal terms is equivalent to increasing the sum of the log eigenvalues. One approach is to set the off-diagonal terms of $\Sigma_Z$ to zero. However, VICReg maximizes the sum of the diagonal term instead of the log of diagonal terms, which is an upper. An exciting research direction is to maximize the eigenvalues of $Z$ using more sophisticated methods, such as using a differential expression for eigendecomposition.

## 4.2 EMPIRICAL EVALUATION

### 4.2.1 VALIDATION OF OUR ASSUMPTIONS

Based on the theory presented in Section 3.3, the conditional output density $p_{\mathbf{z}|x=i}$ reduces to a single Gaussian with decreasing input noise. We validated it using a ResNet-18 model trained with SimCLR or VICReg on the CIFAR-10 dataset (Krizhevsky, 2009). From the test dataset, we sample 512 Gaussian samples for each image and analyzed whether each sample remains Gaussian in the penultimate layer of the DN. Then, we employ the D'Agostino and Pearson's test (D'Agostino, 1971). Figure 1 (left) shows the p-value as a function of the normalized standard deviation. For small noise, we can reject the hypothesis that the conditional output density of the network is not Gaussian (85% for VICReg). Increasing the input noise causes the network's output to become less Gaussian. Although the results indicate that the output of the network is Gaussian, even for the small noise regime, there is a 15% of Type I error.

The next step is to try to confirm our assumption that the model of the data distribution has non-overlapping effective support. We calculate the distribution of pairwise $l_2$ distances between images for seven datasets: MNIST, CIFAR10, CIFAR100, Flowers102, Food101, FGVAircaft. In Figure appendix D, we can see that even for raw pixels, the pairwise distances are far from zero, which means you can use a small Gaussian around each point without overlapping. Therefore, the effective support of these datasets are not-overlapping, and our assumption is realistic.

### 4.2.2 OPTIMIZING THE MUTUAL INFORMATION OBJECTIVE

Implementing Eq 9 in practice requires many "design choices". In section 4.1, we discuss how VICReg uses an approximation of the entropy that is both loose and an upper bound on the true entropy. Next, we suggest combining the VICReg invariance term with different methods for optimizing the entropy.

**Estimators.** The VICReg objective aims to approximate the log determinate of the empirical covariance matrix by using diagonal terms. However, this estimator can be problematic Huber et al. (2008). Instead, we use the LogDet Entropy Estimator Zhouyin & Liu (2021), which provides a tighter upper bound. This estimator is still an upper bound on entropy, which does not provide any guarantee. To address this problem, we also use a lower bound, based on the pairwise distances of the individual Gaussians (Kolchinsky & Tracey, 2017). These proposed methods are compared with recent SSL methods - SimCLR (Chen et al., 2020) and Barlow Twin (Zbontar et al., 2021).

**Setup** Our experiments are conducted on CIFAR-10 Krizhevsky et al. (2009). We use ResNet-18 (He et al., 2016) as our backbone. We use linear evaluation for the quality of the representation. For full details see Appendix E.

**Results.** It can be seen from Table E that the proposed estimators outperform both the original VICReg and SimCLR as well as Barlow Twin. By estimating the entropy with a more accurate estimator, we can improve the results of VICReg, and the pairwise distance estimator, which is a lower bound, achieves the best results. This aligns with the theory that we want to maximize a lower bound on true entropy. The results of our study suggest that a smart selection of entropy estimators, inspired by our framework, leads to better results.

## 5 Self Supervised Learning, EM and Information

Several SSL methods employ the stop gradient operator and only train with positive pairs of data (Grill et al., 2020; Chen & He, 2021). According to (Chen & He, 2021), presetting the stop gradient operation implicitly involves presenting two sets of variables where the algorithm alternates between optimizing each set. Next, we formalize these SSL methods as generalized EM optimization problems, link them to information theory, and analyze how specific design choices affect their collapse.

### 5.1 The EM algorithm and self supervised learning

The classical approach to learning with hidden variables is based on the Expectation Maximization (EM) algorithm (Dempster et al., 1977). Neal & Hinton (1998) showed that we can view it as a dual optimization where both steps are seen as maximizing the same function, $F(\tilde{P}, \theta) = \mathbb{E}_{\tilde{P}}[P(Z, Z'|\theta)] + H(\tilde{P})$ where $H(\tilde{P}) = -\mathbb{E}_{\tilde{P}}\left[\log \tilde{P}(z')\right]$ is the entropy of the empirical distribution $\tilde{P}$ and $\mathbb{E}_{\tilde{P}}[P(Z, Z'|\theta)]$ is the regular likelihood. Using this formulation, Neal & Hinton (1998) showed that the (G)EM algorithm maximizes a variational lower bound on the log likelihood. However, as discussed in section 3.1, optimize the likelihood can be problematic when both variables are changing. Unlike the classic EM algorithm, for SSL, our input variable $Z$ changes in each iteration, and the optimization is with respect to both $Z$ and $Z'$.

### 5.2 Preventing Point Collapse under the EM algorithm

For Gaussian mixture models (GMMs), clustering consists of estimating the parameters that maximize its likelihood function, followed by assigning to each data point the cluster corresponding to its most likely multivariate Gaussian distribution. Chen & He (2021) suggested that the SimSiam method can be viewed as the K-means algorithm, which can be derived by reducing the GMMs

Let us examine a toy dataset on the pattern of two intertwining moons to illustrate the collapse phenomenon under GMM (Figure 1 - right). We begin by training a classical GMM with maximum likelihood, where the means are initialized based on random samples, and the covariance is used as the identity matrix. A red dot represents the Gaussian's mean after training, while a blue dot represents the data points. In the presence of fixed input samples, we observe that there is no collapsing and that the entropy of the centers is high (Figure 4 - left, in the Appendix). However, when we make the input samples trainable and optimize their location, all the points collapse into a single point, resulting in a sharp decrease in entropy (Figure 4 - right, in the Appendix).

To prevent collapse, we follow the K-means algorithm in enforcing sparse posteriors, i.e. using small initial standard deviations and learning only the mean. This forces a one-to-one mapping which leads all points to be closest to the mean without collapsing, resulting in high entropy (Figure 4 - middle, in the Appendix). Another option to prevent collapse is to use different learning rates for input and parameters. Using this setting, the collapsing of the parameters does not maximize the likelihood. Figure 1 (right) shows the results of GMM with different learning rates for learned inputs and parameters. When the parameter learning rate is sufficiently high in comparison to the input learning rate, the entropy decreases much more slowly and no collapse occurs.

## 6 Benefits of Information Maximization for Generalization

The purpose of this section is to further connect the invariance loss, the covariance matrix, and the information with the input to the generalization ability of the model by deriving a novel generalization bound. Together with the results from the previous sections, this provides a mathematical understanding of the benefits of SSL through maximization of information with implicit regularization.

### 6.1 Notation

Let $x$ be our input and $y \in \mathbb{R}^r$ the output. We are given a labeled training data $S = ((x_i, y_i))_{i=1}^n$ of size $n$ and an unlabeled training data $\bar{S} = ((x_i^+, x_i^{++}))_{i=1}^m$ of size $m$, where $x_i^+$ and $x_i^{++}$ share the same (unknown) label. With the unlabeled training data, we define the invariance loss $I_{\bar{S}}(f_\theta) = \frac{1}{m}\sum_{i=1}^m \|f_\theta(x_i^+) - f_\theta(x_i^{++})\|$ where $f_\theta$ is the trained representation on the unlabeled

data $\bar{S}$. We define a labeled loss $\ell_{x,y}(w) = \|Wf_\theta(x) - y\|$ where $w = \text{vec}[W] \in \mathbb{R}^{dr}$ is the vectorization of the matrix $W \in \mathbb{R}^{r \times d}$. Let $w_S = \text{vec}[W_S]$ be the minimum norm solution as $W_S = \text{minimize}_{W'} \|W'\|_F$ s.t. $W' \in \arg\min_W \frac{1}{n} \sum_{i=1}^n \|Wf_\theta(x_i) - y_i\|^2$. We also define the representation matrices $Z_S = [f(x_1), \ldots, f(x_n)] \in \mathbb{R}^{d \times n}$ and $Z_{\bar{S}} = [f(x_1^+), \ldots, f(x_m^+)] \in \mathbb{R}^{d \times m}$, and the projection matrices $\mathbf{P}_{Z_S} = I - Z_S^\top (Z_S Z_S^\top)^\dagger Z_S$ and $\mathbf{P}_{Z_{\bar{S}}} = I - Z_{\bar{S}}^\top (Z_{\bar{S}} Z_{\bar{S}}^\top)^\dagger Z_{\bar{S}}$. We define the label matrix $Y_S = [y_1, \ldots, y_n]^\top \in \mathbb{R}^{n \times r}$ and the unknown label matrix $Y_{\bar{S}} = [y_1^+, \ldots, y_m^+]^\top \in \mathbb{R}^{m \times r}$, where $y_i^+$ is the unknown label of $x_i^+$. Let $\mathcal{F}$ be a hypothesis space of $f_\theta$. For a given hypothesis space $\mathcal{F}$, we define the normalized Rademacher complexity $\hat{\mathcal{R}}_m(\mathcal{F}) = \frac{1}{\sqrt{m}} \mathbb{E}_{\bar{S},\xi}[\sup_{f \in \mathcal{F}} \sum_{i=1}^m \xi_i \|f(x_i^+) - f(x_i^{++})\|]$, where , $\xi_1, \ldots, \xi_m$ are independent uniform random variables taking values in $\{-1, 1\}$. It is normalized such that $\tilde{\mathcal{R}}_m(\mathcal{F}) = O(1)$ as $m \to \infty$ for typical choices of hypothesis spaces $\mathcal{F}$, including DNs (Bartlett et al., 2017; Kawaguchi et al., 2018).

## 6.2 Generalization Bound for VICReg

We now show that SSL via VICReg can be understood to improve the generalization ability for the supervised downstream task. Namely, Theorem 2 shows that the expected labeled loss $\mathbb{E}_{x,y}[\ell_{x,y}(w_S)]$ is minimized when we minimize the unlabeled invariance loss $I_{\bar{S}}(f_\theta)$ while controlling the covariance $Z_{\bar{S}} Z_{\bar{S}}^\top$ and the complexity of representations $\tilde{\mathcal{R}}_m(\mathcal{F})$:

**Theorem 2.** *(Informal version). For any $\delta > 0$, with probability at least $1 - \delta$, the following holds:*

$$\mathbb{E}_{x,y}[\ell_{x,y}(w_S)] \leq I_{\bar{S}}(f_\theta) + \frac{2}{\sqrt{m}} \|\mathbf{P}_{Z_{\bar{S}}} Y_{\bar{S}}\|_F + \frac{1}{\sqrt{n}} \|\mathbf{P}_{Z_S} Y_S\|_F + \frac{2\tilde{\mathcal{R}}_m(\mathcal{F})}{\sqrt{m}} + \mathcal{Q}_{m,n}, \quad (13)$$

*where $\mathcal{Q}_{m,n} = O(G\sqrt{\frac{\ln(1/\delta)}{m}} + \sqrt{\frac{\ln(1/\delta)}{n}}) \to 0$ as $m, n \to \infty$. In $\mathcal{Q}_{m,n}$, the value of $G$ for the term decaying at the rate $1/\sqrt{m}$ depends on the hypothesis space of $f_\theta$ and $w$ whereas the term decaying at the rate $1/\sqrt{n}$ is independent of any hypothesis space.*

*Proof.* The complete version of Theorem 2 and its proof are presented in Appendix G. $\qquad\square$

Note that our framework holds for a classification with a linear layer and the $l_2$ norm as the loss. Also, in order that this bounds will not become vacuous we should imposed that the class $\mathcal{F}$ has a finite norm range and that the class of matrices for the linear layer $\mathcal{W}$ is of finite norm.

The term $\|\mathbf{P}_{Z_{\bar{S}}} Y_{\bar{S}}\|_F$ in Theorem 2 contains the unobservable label matrix $Y_{\bar{S}}$. However, we can minimize this term by using $\|\mathbf{P}_{Z_{\bar{S}}} Y_{\bar{S}}\|_F \leq \|\mathbf{P}_{Z_{\bar{S}}}\|_F \|Y_{\bar{S}}\|_F$ and by minimizing $\|\mathbf{P}_{Z_{\bar{S}}}\|_F$. The factor $\|\mathbf{P}_{Z_{\bar{S}}}\|_F$ is minimized when the rank of the covariance $Z_{\bar{S}} Z_{\bar{S}}^\top$ is maximized. Since a strictly diagonally dominant matrix is non-singular, this can be enforced by maximizing the diagonal entries while minimizing the off-diagonal entries, as is done in VICReg. For example, if $d \geq n$, then $\|\mathbf{P}_{Z_{\bar{S}}}\|_F = 0$ when the covariance $Z_{\bar{S}} Z_{\bar{S}}^\top$ is of full rank. The term $\|\mathbf{P}_{Z_S} Y_S\|_F$ contains only observable variables and we can directly measure the value of this term using training data. In addition, the term $\|\mathbf{P}_{Z_S} Y_S\|_F$ is also minimized when the rank of the covariance $Z_S Z_S^\top$ is maximized. Since the covariances $Z_S Z_S^\top$ and $Z_{\bar{S}} Z_{\bar{S}}^\top$ concentrate to each other via concentration inequalities with the error in the order of $O(\sqrt{(\ln(1/\delta))/n} + \tilde{\mathcal{R}}_m(\mathcal{F})\sqrt{(\ln(1/\delta))/m})$, we can also minimize the upper bound on $\|\mathbf{P}_{Z_S} Y_S\|_F$ by maximizing the diagonal entries of $Z_{\bar{S}} Z_{\bar{S}}^\top$ while minimizing its off-diagonal entries, as is done in VICReg.

Thus, VICReg can be understood as a method to minimize the generalization bound in Theorem 2 by minimizing the invariance loss while controlling the covariance $Z_{\bar{S}} Z_{\bar{S}}^\top$ to minimize the *label-agnostic* upper bounds on $\|\mathbf{P}_{Z_{\bar{S}}} Y_{\bar{S}}\|_F$ and $\|\mathbf{P}_{Z_S} Y_S\|_F$. If we know *partial* information about the label $Y_{\bar{S}}$ of the unlabeled data, we can use it to minimize $\|\mathbf{P}_{Z_{\bar{S}}} Y_{\bar{S}}\|_F$ and $\|\mathbf{P}_{Z_S} Y_S\|_F$ directly. This direction can be used to improve VICReg in future work for the partial observable setting.

## 6.3 Understanding via Mutual Information

Theorem 2 together with the result of the previous section shows that, for generalization in the downstream task, it is helpful to maximize the mutual information $I(Z; X')$ in SSL via minimizing

the invariance loss $I_{\bar{S}}(f_\theta)$ while controlling the covariance $Z_{\bar{S}}Z_{\bar{S}}^\top$. The term $\frac{2\tilde{\mathcal{R}}_m(\mathcal{F})}{\sqrt{m}}$ captures the importance of controlling the complexity of the representations $f_\theta$. To understand this term further in terms of mutual information, let us consider a discretization of the parameter space of $\mathcal{F}$ to have finite $|\mathcal{F}| < \infty$ (indeed, a computer always implements some discretization of continuous variables). Then, by Massart's Finite Class Lemma, we have that $\tilde{\mathcal{R}}_m(\mathcal{F}) \leq C\sqrt{\ln|\mathcal{F}|}$ for some constant $C > 0$. Moreover, Shwartz-Ziv (2022) shows that we can approximate $\ln|\mathcal{F}|$ by $2^{I(Z;X)}$. Thus, in Theorem 2, the term $I_{\bar{S}}(f_\theta) + \frac{2}{\sqrt{m}}\|\mathbf{P}_{Z_{\bar{S}}}Y_{\bar{S}}\|_F + \frac{1}{\sqrt{n}}\|\mathbf{P}_{Z_S}Y_S\|_F$ corresponds to $I(Z;X')$ while the term of $\frac{2\tilde{\mathcal{R}}_m(\mathcal{F})}{\sqrt{m}}$ corresponds to $I(Z;X)$. Recall that the information can be decomposed as

$$I(Z;X) = I(Z;X') + I(Z;X|X').\tag{14}$$

where we want to maximize the predictive information $I(Z;X')$, while minimizing $I(Z;X)$ (**??**). Thus, in order to improve generalization, we also need to control $\frac{2\tilde{\mathcal{R}}_m(\mathcal{F})}{\sqrt{m}}$ to restrict the superfluous information $I(Z;X|X')$, in addition to minimize $I_{\bar{S}}(f_\theta) + \frac{2}{\sqrt{m}}\|\mathbf{P}_{Z_{\bar{S}}}Y_{\bar{S}}\|_F + \frac{1}{\sqrt{n}}\|\mathbf{P}_{Z_S}Y_S\|_F$ that corresponded to maximize the predictive information $I(Z;X')$. Although we can explicitly add regularization on $I(Z;X|X')$ to control $\frac{2\tilde{\mathcal{R}}_m(\mathcal{F})}{\sqrt{m}}$, it is possible that $I(Z;X|X')$ and $\frac{2\tilde{\mathcal{R}}_m(\mathcal{F})}{\sqrt{m}}$ are implicitly regularized via implicit bias through e design choises (Gunasekar et al., 2017; Soudry et al., 2018; Gunasekar et al., 2018). Thus, Theorem 2 connects the information-theoretic understanding of SSL with the probabilistic guarantee on the generalization ability.

## 6.4 COMPARING GENERALIZATION BOUNDS

The generalization bound of SimCLR (Saunshi et al., 2019) requires the number of label classes to go infinity to make the generalization gap decrease towards zero. In contrast, the bound on VICReg in Theorem 2 does *not* require the number of label classes to approach infinity to let the generalization gap go to zero. This reflects the fact that, unlike SimCLR, VICReg does not use negative pairs and thus does not use a loss function that is based on the implicit expectation that the labels of a negative pair $(y^+, y^-)$ are different. Another difference is that our VICReg bound improves as $n$ increases, while the previous bound of SimCLR (Saunshi et al., 2019) does not depend on $n$. This is because the previous work assumes partial access to the true distribution of $x$ given $y$ per class for setting $W$, which removes the importance of labeled data size $n$ and is not assumed in our study. Consequently, our bound provides a new insight for VICReg regarding the ratio of the effects of $m$ v.s. $n$ through $G\sqrt{\ln(1/\delta)/m} + \sqrt{\ln(1/\delta)/n}$. Finally, Theorem 2 also illuminates the advantages of VICReg over standard supervised training. That is, with standard training, the generalization bound via the Rademacher complexity requires the complexities of hypothesis spaces, $\tilde{\mathcal{R}}_n(\mathcal{W})/\sqrt{n}$ and $\tilde{\mathcal{R}}_n(\mathcal{F})/\sqrt{n}$, with respect to the size of labeled data $n$, instead of the size of unlabeled data $m$. Here, $\tilde{\mathcal{R}}_n(\mathcal{W})$ is the the normalized Rademacher complexity for the hypothesis space of $w$. Thus, Theorem 2 shows that using SSL, we can replace all the complexities of hypothesis spaces in terms of $n$ with those in terms of $m$. Since $m$ is typically much larger than $n$, this illuminates the benefit of SSL.

## 7 CONCLUSIONS

In this study, we examine SSL's objective function from an information-theoretic perspective. Based on transfering of the required stochasticity to the input distribution, we show how SSL objectives can be derived. Thus, even when using deterministic DNs, it is possible to perform an information-theoretic analysis. The second part of the paper rediscovered SSL loss functions from first principles and demonstrated their implicit assumptions. We empirically validated our analysis and confirmed the validity of our novel understanding. As a result of our analysis, we have proposed new SSL algorithms that perform better than existing ones. Furthermore, we derived a generalization bound on the downstream task, tight it to known information objeective terms and demonstrate that VICReg minimizes it. In addition, our work opens many new avenues for future research, including a better estimation of information-theoretic quantities that are consistent with our assumptions and identifying which SSL method is the most appropriate according to data characteristics. In addition, our probabilistic guarantee suggests that VICReg can be further improved for the setting of partial label information by aligning the covariance matrix with the partially observable label matrix.

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

# A DATA DISTRIBUTION AFTER DEEP NETWORK TRANSFORMATION

**Theorem 3.** *Given the setting of eq. (4) the unconditional DN output density denoted as $Z$ approximates (given the truncation of the Gaussian on its effective support that is included within a single region $\omega$ of the DN's input space partition) a mixture of the affinely transformed distributions $x|x^*_{n(x)}$ e.g. for the Gaussian case*

$$Z \sim \sum_{n=1}^{N} \mathcal{N}\Big(\boldsymbol{A}_{\omega(\boldsymbol{x}^*_n)}\boldsymbol{x}^*_n + \boldsymbol{b}_{\omega(\boldsymbol{x}^*_n)}, \boldsymbol{A}^T_{\omega(\boldsymbol{x}^*_n)}\Sigma_{\boldsymbol{x}^*_n}\boldsymbol{A}_{\omega(\boldsymbol{x}^*_n)}\Big)^{T=n},$$

*where $\omega(\boldsymbol{x}^*_n) = \omega \in \Omega \iff \boldsymbol{x}^*_n \in \omega$ is the partition region in which the prototype $\boldsymbol{x}^*_n$ lives in.*

*Proof.* We know that If $\int_\omega p(\boldsymbol{x}|\boldsymbol{x}^*_{n(\boldsymbol{x})})d\boldsymbol{x} \approx 1$ then $f$ is linear within the effective support of $p$. Therefore, any sample from $p$ will almost surely lie within a single region $\omega \in \Omega$ and therefore the entire mapping can be considered linear with respect to $p$. Thus, the output distribution is a linear transformation of the input distribution based on the per-region affine mapping. $\square$

# B  LOWER BOUNDS ON $\mathbb{E}_{x'}\left[\log q(z|x')\right]$

In this appendix we present the full derivation of the lower bound on $\mathbb{E}_{x'}\left[\log q(z|x')\right]$.

Because $Z'|X'$ is a Gaussian, we can write it as $Z' = \mu(x') + L(x')\epsilon$ where $\epsilon \sim \mathcal{N}(0,1)$ and $L(x')^T L(x') = \Sigma(x')$. Now, setting $\Sigma_r = I$, will give us:

$$\mathbb{E}_{x'}\left[\log q(z|x')\right] \geq \tag{15}$$

$$\mathbb{E}_{z'|x'}\left[\log q(z|z')\right] = \tag{16}$$

$$\mathbb{E}_{z'|x'}\left[\frac{d}{2}\log 2\pi - \frac{1}{2}(z-z')^T (I))^{-1}(z-z')\right] = \tag{17}$$

$$\frac{d}{2}\log 2\pi - \frac{1}{2}\mathbb{E}_{z'|x'}\left[(z-z')^2\right] = \tag{18}$$

$$\frac{d}{2}\log 2\pi - \frac{1}{2}\mathbb{E}_\epsilon\left[(z-\mu(x')-L(x')\epsilon)^2\right] = \tag{19}$$

$$\frac{d}{2}\log 2\pi - \frac{1}{2}\mathbb{E}_\epsilon\left[(z-\mu(x'))^2 - 2(z-\mu(x')*L(x')\epsilon) + \left((L(x')\epsilon)^T(L(x')\epsilon)\right)\right] = \tag{20}$$

$$\frac{d}{2}\log 2\pi - \frac{1}{2}\mathbb{E}_\epsilon\left[(z-\mu(x'))^2\right] + (z-\mu(x')L(x'))\mathbb{E}_\epsilon\left[\epsilon\right] - \frac{1}{2}\mathbb{E}_\epsilon\left[\epsilon^T L(x')^T L(x')\epsilon\right] = \tag{21}$$

$$\frac{d}{2}\log 2\pi - \frac{1}{2}(z-\mu(x'))^2 - \frac{1}{2}Tr\log\Sigma(x') \tag{22}$$

where $\mathbb{E}_{x'}\left[\log q(z|x')\right] = \mathbb{E}_{x'}\left[\log\mathbb{E}_{z'|x'}\left[q(z|z')\right]\right] \geq \mathbb{E}_{z'}\left[\log q(z|z')\right]$ by Jensen's inequality, $\mathbb{E}_\epsilon[\epsilon] = 0$ and $\mathbb{E}_\epsilon\left[\epsilon\left(L(x')^T L(x')\epsilon\right)\right] = Tr\log\Sigma(x')$ by the Hutchinson's estimator.

$$\mathbb{E}_{z|x}\left[\mathbb{E}_{z'|x'}\left[\log q(z|z')\right]\right] = \tag{23}$$

$$\mathbb{E}_{z|x}\left[\frac{d}{2}\log 2\pi - \frac{1}{2}(z-\mu(x'))^2 - \frac{1}{2}Tr\log\Sigma(x')\right] = \tag{24}$$

$$\frac{d}{2}\log 2\pi - \frac{1}{2}\mathbb{E}_{z|x}\left[(z-\mu(x'))^2\right] - \frac{1}{2}Tr\log\Sigma(x') = \tag{25}$$

$$\frac{d}{2}\log 2\pi - \frac{1}{2}\mathbb{E}_\epsilon\left[(\mu(x)+L(x)\epsilon-\mu(x'))^2\right] - \frac{1}{2}Tr\log\Sigma(x') = \tag{26}$$

$$\frac{d}{2}\log 2\pi - \frac{1}{2}\mathbb{E}_\epsilon\left[(\mu(x)-\mu(x'))^2\right] + \mathbb{E}_\epsilon\left[(\mu(x)-\mu(x'))L(x)\epsilon\right] \tag{27}$$

$$- \frac{1}{2}\mathbb{E}_\epsilon\left[\epsilon^T L(x)^T L(x)\epsilon\right] - \frac{1}{2}Tr\log\Sigma(x') = \tag{28}$$

$$\frac{d}{2}\log 2\pi - \frac{1}{2}(\mu(x)-\mu(x'))^2 - \frac{1}{2}Tr\log\Sigma(x) - \frac{1}{2}Tr\log\Sigma(x') = \tag{29}$$

$$\frac{d}{2}\log 2\pi - \frac{1}{2}(\mu(x)-\mu(x'))^2 - \frac{1}{2}\log\left(|\Sigma(x)|\cdot|\Sigma(x')|\right) \tag{30}$$

## C  ENTROPY ESTIMATORS

The estimation of entropy is one of the classic problems in information theory, where Gaussian mixture density is one of the most popular representations. With a sufficient number of components, they can approximate any smooth function with arbitrary accuracy. For Gaussian mixtures, there is, however, no closed-form solution to differential entropy. There exist several approximations in the literature, including loose upper and lower bounds (Huber et al., 2008). Monte Carlo (MC) sampling is one way to approximate Gaussian mixture entropy. With sufficient MC samples, an unbiased estimate of entropy with an arbitrarily accurate can be obtained. Unfortunately, MC sampling is a very computationally expensive and typically requires a large number of samples, especially in high dimensions (Brewer, 2017). Using the first two moments of the empirical distribution, VIGCreg used one of the most straightforward approaches for approximating the entropy. Despite this, previous studies have found that this method is a poor approximation of the entropy in many cases Huber et al. (2008). Another options is to use the LogDet function. Several estimators have been proposed to implement it, including uniformly minimum variance unbiased (UMVU) (Ahmed & Gokhale, 1989), and bayesian methods Misra et al. (2005). These methods, however, often require complex optimizations. The LogDet estimator presented in Zhouyin & Liu (2021) used the differential entropy $\alpha$ order entropy using scaled noise. They demonstrated that it can be applied to high-dimensional features and is robust to random noise. Based on Taylor-series expansions, Huber et al. (2008) presented a lower bound for the entropy of Gaussian mixture random vectors. They use Taylor-series expansions of the logarithm of each Gaussian mixture component to get an analytical evaluation of the entropy measure. In addition, they present a technique for splitting Gaussian densities to avoid components with high variance, which would require computationally expensive calculations. Kolchinsky & Tracey (2017) introduce a novel family of estimators for the mixture entropy. For this family, a pairwise-distance function between component densities defined for each member. These estimators are computationally efficient, as long as the pairwise-distance function and the entropy of each component distribution are easy to compute. Moreover, the estimator is continuous and smooth and is therefore useful for optimization problems. In addition, they presented both lower bound (using Chernoff distance) and an upper bound (using the KL divergence) on the entropy, which are are exact when the component distributions are grouped into well-separated clusters,

## D  EMPIRICAL VALIDATION OF OUR ASSUMPTION

We will try to verify empirically our assumptions on different datasets We compute the pairwise $l2$ distances between images for seven datasets: MNIST, CIFAR10, CIFAR100, Flowers102, Food101, and FGVAircaft. We found that even for raw pixels, the pairwise distances are far from zero, which means you can use a small Gaussian around each point without overlapping. Consequently, the effective supports of these high-dimensional datasets are not overlapping, and our assumption is realistic even for current popular SSL datasets..

## E  EXPERIMENTAL VERIFICATION OF INFORMATION-BASED BOUND OPTIMIZATION

**Setup** Our experiments are conducted on CIFAR-10 Krizhevsky et al. (2009). We use ResNet-18 (He et al., 2016) as our backbone. Each model is trained with $512$ batch size for $800$ epochs. We use linear evaluation to assess the quality of the representation. Once the model has been pre-trained, we follow the same fine-tuning procedures as for the baseline methods (Caron et al., 2020).

## F  EXPECTATION MAXIMIZATION AND COLLAPSING

## G  ON BENEFITS OF INFORMATION MAXIMIZATION FOR GENERALIZATION

In this Appendix, we present the complete version of Theorem 2 along with its proof and additional discussions.

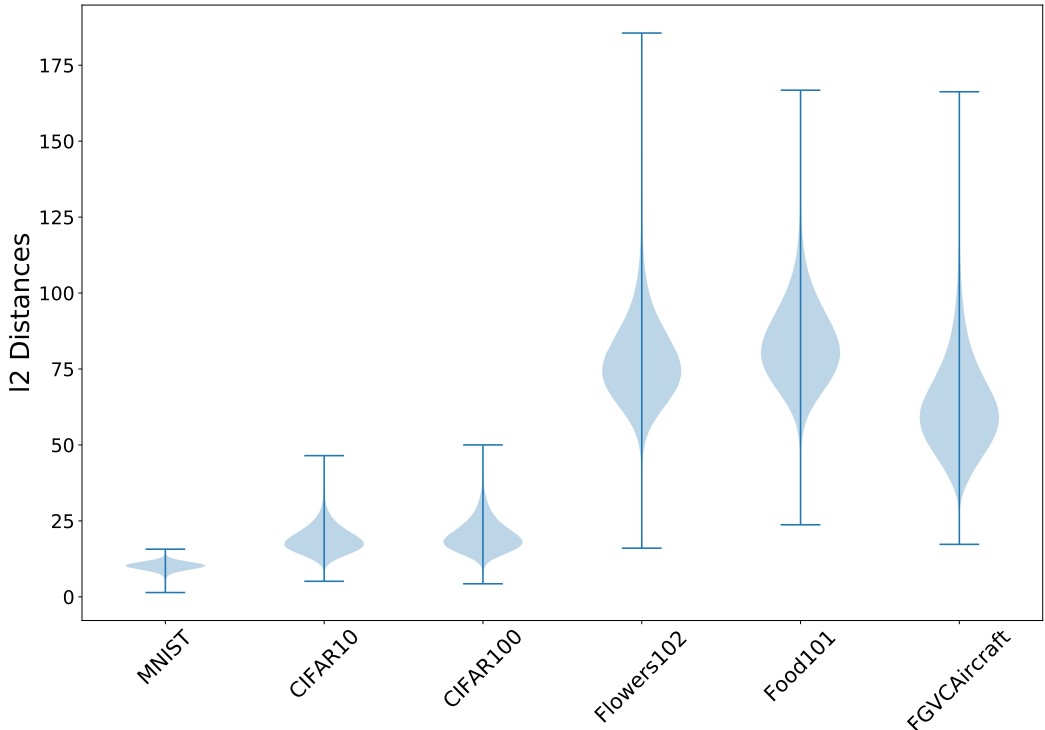

Figure 2: **The Gaussians around each point are not overlapping** The $l2$ distances between raw images for different datasets

| Method | Accuracy |
|---|---|
| SimCLR | $89.72 \pm 0.05$ |
| Barlow Twins | $88.81 \pm 0.1$ |
| VICReg | $89.32 \pm 0.09$ |
| VICReg + Pairwise Distances Estimator (ours) | $90.09 \pm 0.09$ |
| VICReg + Log Derminate Estimator (ours) | $89.77 \pm 0.08$ |

Table 1: **Entropy estimator achieved better results on SSL** - CIFAR10 accuracy on linear evaluation of SSL for different entropy estimators. The best results achieved by pairwise distances lower bound

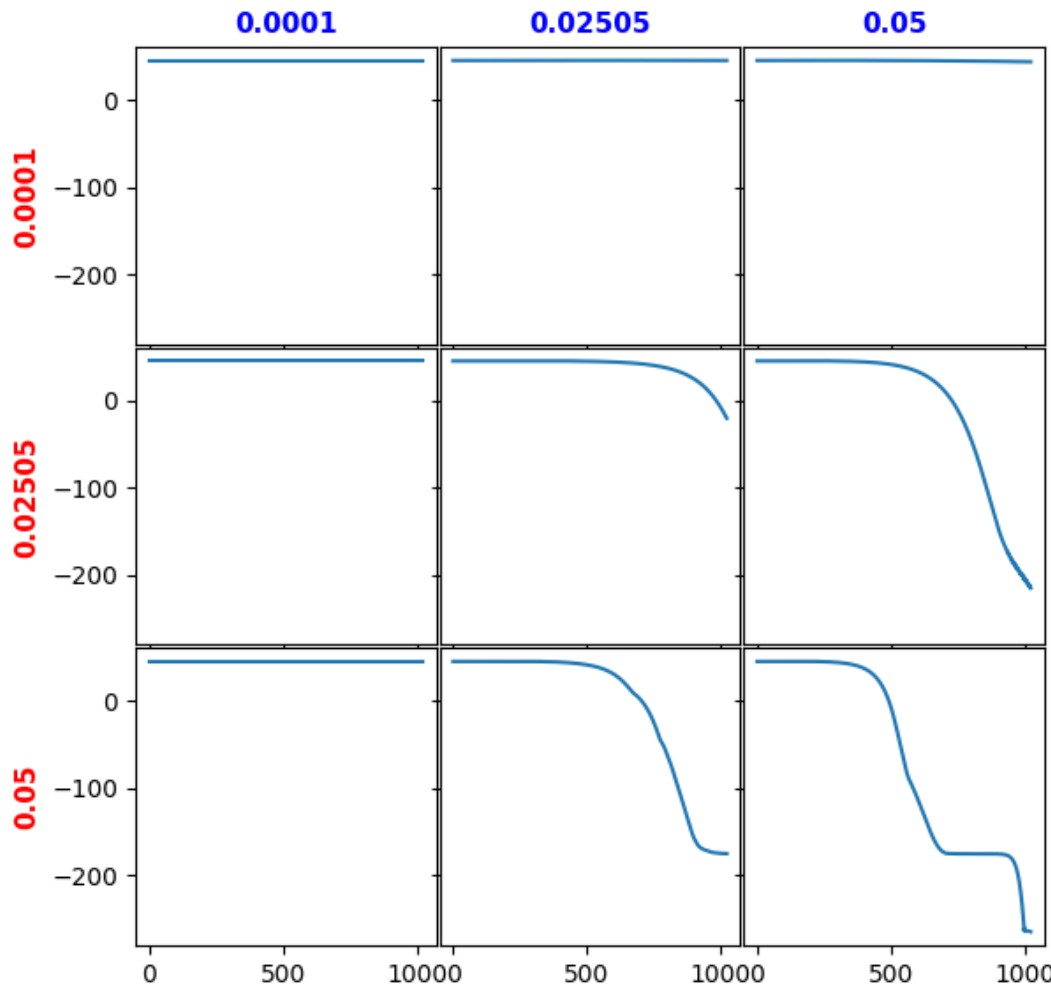

Figure 3: Evolution of the entropy for each of the learning rate configurations showing that the impact of picking the incorrect learning rate for the data and/or centroids lead to a collapse of the samples.

### G.1 ADDITIONAL NOTATION AND DETAILS

We start to introduce additional notation and details. We use the notation of $x \in \mathcal{X}$ for an input and $y \in \mathcal{Y} \subseteq \mathbb{R}^r$ for an output. Define $p(y) = \mathbb{P}(Y = y)$ to be the probability of getting label $y$ and $\hat{p}(y) = \frac{1}{n} \sum_{i=1}^n \mathbb{1}\{y_i = y\}$ to be the empirical estimate of $p(y)$. Let $\zeta$ be an upper bound on the norm of the label as $\|y\|_2 \leq \zeta$ for all $y \in \mathcal{Y}$. Define the minimum norm solution $W_{\bar{S}}$ of the unlabeled data as $W_{\bar{S}} = \text{minimize}_{W'} \|W'\|_F$ s.t. $W' \in \arg\min_W \frac{1}{m} \sum_{i=1}^m \|Wf_\theta(x_i^+) - g^*(x_i^+)\|^2$. Let $\kappa_S$ be a data-dependent upper bound on the per-sample Euclidian norm loss with the trained model as $\|W_S f_\theta(x) - y\| \leq \kappa_S$ for all $(x, y) \in \mathcal{X} \times \mathcal{Y}$. Similarly, let $\kappa_{\bar{S}}$ be a data-dependent upper bound on the per-sample Euclidian norm loss as $\|W_{\bar{S}} f_\theta(x) - y\| \leq \kappa_{\bar{S}}$ for all $(x, y) \in \mathcal{X} \times \mathcal{Y}$. Define the difference between $W_S$ and $W_{\bar{S}}$ by $c = \|W_S - W_{\bar{S}}\|_2$. Let $\mathcal{W}$ be a hypothesis space of $W$ such that $W_{\bar{S}} \in \mathcal{W}$. We denote by $\tilde{\mathcal{R}}_m(\mathcal{W} \circ \mathcal{F}) = \frac{1}{\sqrt{m}} \mathbb{E}_{\bar{S},\xi}[\sup_{W \in \mathcal{W}, f \in \mathcal{F}} \sum_{i=1}^m \xi_i \|g^*(x_i^+) - Wf(x_i^+)\|]$ the normalized Rademacher complexity of the set $\{x^+ \mapsto \|g^*(x^+) - Wf(x^+)\| : W \in \mathcal{W}, f \in \mathcal{F}\}$. we denote by $\kappa$ a upper bound on the per-sample Euclidian norm loss as $\|Wf(x) - y\| \leq \kappa$ for all $(x, y, W, f) \in \mathcal{X} \times \mathcal{Y} \times \mathcal{W} \times \mathcal{F}$.

We adopt the following data-generating process model that is used in the previous paper on analyzing contrastive learning (Saunshi et al., 2019). For the labeled data, first, $y$ is drawn from the distritbuion

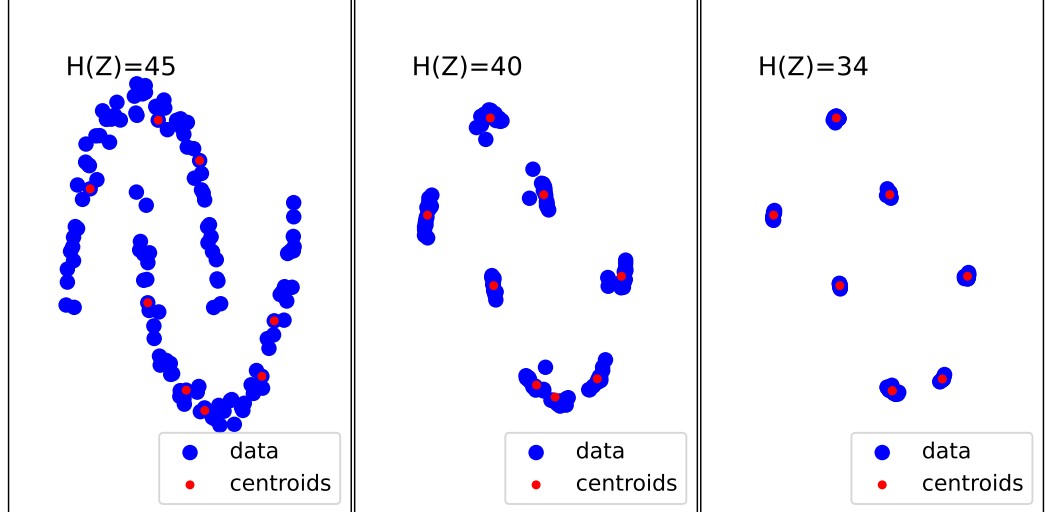

Figure 4: **Evolution of GMM training when enforcing a one-to-one mapping between the data and centroids akin to K-means i.e. using a small and fixed covariance matrix. We see that collapse does not occur.** Left - In the presence of fixed input samples, we observe that there is no collapsing and that the entropy of the centers is high. Right - when we make the input samples trainable and optimize their location, all the points collapse into a single point, resulting in a sharp decrease in entropy.

$\rho$ on $\mathcal{Y}$, and then $x$ is drawn from the conditional distribution $\mathcal{D}_y$ conditioned on the label $y$. That is, we have the join distribution $\mathcal{D}(x, y) = \mathcal{D}_y(x)\rho(y)$ with $((x_i, y_i))_{i=1}^n \sim \mathcal{D}^n$. For the unlabeled data, first, each of the *unknown* labels $y^+$ and $y^-$ is drawn from the distritbuion $\rho$, and then each of the positive examples $x^+$ and $x^{++}$ is drawn from the conditional distribution $\mathcal{D}_{y^+}$ while the negative example $x^-$ is drawn from the $\mathcal{D}_{y^-}$. Unlike the analysis of contrastive learning, we do not require the negative samples. Let $\tau_{\bar{S}}$ be a data-dependent upper bound on the invariance loss with the trained representation as $\|f_\theta(\bar{x}) - f_\theta(x)\| \le \tau_{\bar{S}}$ for all $(\bar{x}, x) \sim \mathcal{D}_y^2$ and $y \in \mathcal{Y}$. Let $\tau$ be a data-independent upper bound on the invariance loss with the trained representation as $\|f(\bar{x}) - f(x)\| \le \tau$ for all $(\bar{x}, x) \sim \mathcal{D}_y^2$, $y \in \mathcal{Y}$, and $f \in \mathcal{F}$. For the simplicity, we assume that there exists a function $g^*$ such that $y = g^*(x) \in \mathbb{R}^r$ for all $(x, y) \in \mathcal{X} \times \mathcal{Y}$. Discarding this assumption adds the average of label noises to the final result, which goes to zero as the sample sizes $n$ and $m$ increase, assuming that the mean of the label noise is zero.

## G.2 RESULT

The following theorem is the complete version of Theorem 2:

**Theorem 4.** *For any $\delta > 0$, with probability at least $1 - \delta$, the following holds:*

$$\mathbb{E}_{x,y}[\ell_{x,y}(w_S)] \le cI_{\bar{S}}(f_\theta) + \frac{2}{\sqrt{m}}\|\mathbf{P}_{Z_{\bar{S}}}Y_{\bar{S}}\|_F + \frac{1}{\sqrt{n}}\|\mathbf{P}_{Z_S}Y_S\|_F + Q_{m,n}, \qquad (31)$$

*where*

$$Q_{m,n} = c\left(\frac{2\tilde{\mathcal{R}}_m(\mathcal{F})}{\sqrt{m}} + \tau\sqrt{\frac{\ln(3/\delta)}{2m}} + \tau_{\bar{S}}\sqrt{\frac{\ln(3/\delta)}{2n}}\right)$$

$$+ \kappa_S\sqrt{\frac{2\ln(6|\mathcal{Y}|/\delta)}{2n}}\sum_{y \in \mathcal{Y}}\left(\sqrt{\hat{p}(y)} + \sqrt{p(y)}\right)$$

$$+ \frac{4\mathcal{R}_m(\mathcal{W} \circ \mathcal{F})}{\sqrt{m}} + 2\kappa\sqrt{\frac{\ln(4/\delta)}{2m}} + 2\kappa_{\bar{S}}\sqrt{\frac{\ln(4/\delta)}{2n}}.$$

*Proof.* The complete proof is presented in Appendix G.3. □

The bound in the complete version of Theorem 4 is better than the one in the informal version of Theorem 2, because of the factor $c$. The factor $c$ measures the difference between the minimum norm solution $W_S$ of the labeled training data and the minimum norm solution $W_{\bar{S}}$ of the unlabeled training data. Thus, the factor $c$ also decreases towards zero as $n$ and $m$ increase. Moreover, if the labeled and unlabeled training data are similar, the value of $c$ is small, decreasing the generalization bound further, which makes sense. Thus, we can view the factor $c$ as a measure on the distance between the labeled training data and the unlabeled training data.

We obtain the informal version from the complete version of Theorem 2 by the following reasoning to simplify the notation in the main text. We have that $cI_{\bar{S}}(f_\theta) + c\frac{2\tilde{\mathcal{R}}_m(\mathcal{F})}{\sqrt{m}} = I_{\bar{S}}(f_\theta) + \frac{2\tilde{\mathcal{R}}_m(\mathcal{F})}{\sqrt{m}} + Q$, where $Q = (c-1)(I_{\bar{S}}(f_\theta) + \frac{2\tilde{\mathcal{R}}_m(\mathcal{F})}{\sqrt{m}}) \le \varsigma \to 0$ as as $m, n \to \infty$, since $c \to 0$ as $m, n \to \infty$. However, this reasoning is used only to simplify the notation in the main text. The bound in the complete version of Theorem 2 is more accurate and indeed tighter than the one in the informal version.

In Theorem 2, $Q_{m,n} \to 0$ as $m, n \to \infty$ if $\frac{\tilde{\mathcal{R}}_m(\mathcal{F})}{\sqrt{m}} \to 0$ as $m \to \infty$. Indeed, this typically holds because $\tilde{\mathcal{R}}_m(\mathcal{F}) = O(1)$ as $m \to \infty$ for typical choices of $\mathcal{F}$, including deep neural networks (Bartlett et al., 2017; Kawaguchi et al., 2018; Golowich et al., 2018) as well as other common machine learning models (Bartlett & Mendelson, 2002; Mohri et al., 2012; Shalev-Shwartz & Ben-David, 2014).

### G.3 Proof of Theorem 2

*Proof of Theorem 2.* Let $W = W_S$ where $W_S$ is the the minimum norm solution as $W_S = \text{minimize}_{W'} \|W'\|_F$ s.t. $W' \in \arg\min_W \frac{1}{n}\sum_{i=1}^n \|Wf_\theta(x_i) - y_i\|^2$. Let $W^* = W_{\bar{S}}$ where $W_{\bar{S}}$ is the minimum norm solution as $W^* = W_{\bar{S}} = \text{minimize}_{W'} \|W'\|_F$ s.t. $W' \in \arg\min_W \frac{1}{m}\sum_{i=1}^m \|Wf_\theta(x_i^+) - g^*(x_i^+)\|^2$. Since $y = g^*(x)$,

$$y = g^*(x) \pm W^*f_\theta(x) = W^*f_\theta(x) + (g^*(x) - W^*f_\theta(x)) = W^*f_\theta(x) + \varphi(x)$$

where $\varphi(x) = g^*(x) - W^*f_\theta(x)$. Define $L_S(w) = \frac{1}{n}\sum_{i=1}^n \|Wf_\theta(x_i) - y_i\|$. Using these,

$$L_S(w) = \frac{1}{n}\sum_{i=1}^n \|Wf_\theta(x_i) - y_i\|$$

$$= \frac{1}{n}\sum_{i=1}^n \|Wf_\theta(x_i) - W^*f_\theta(x_i) - \varphi(x_i)\|$$

$$\ge \frac{1}{n}\sum_{i=1}^n \|Wf_\theta(x_i) - W^*f_\theta(x_i)\| - \frac{1}{n}\sum_{i=1}^n \|\varphi(x_i)\|$$

$$= \frac{1}{n}\sum_{i=1}^n \|\tilde{W}f_\theta(x_i)\| - \frac{1}{n}\sum_{i=1}^n \|\varphi(x_i)\|$$

where $\tilde{W} = W - W^*$. We now consider new fresh samples $\bar{x}_i \sim \mathcal{D}_{y_i}$ for $i = 1, \ldots, n$ to rewrite the above further as:

$$L_S(w) \ge \frac{1}{n}\sum_{i=1}^n \|\tilde{W}f_\theta(x_i) \pm \tilde{W}f_\theta(\bar{x}_i)\| - \frac{1}{n}\sum_{i=1}^n \|\varphi(x_i)\|$$

$$= \frac{1}{n}\sum_{i=1}^n \|\tilde{W}f_\theta(\bar{x}_i) - (\tilde{W}f_\theta(\bar{x}_i) - \tilde{W}f_\theta(x_i))\| - \frac{1}{n}\sum_{i=1}^n \|\varphi(x_i)\|$$

$$\ge \frac{1}{n}\sum_{i=1}^n \|\tilde{W}f_\theta(\bar{x}_i)\| - \frac{1}{n}\sum_{i=1}^n \|\tilde{W}f_\theta(\bar{x}_i) - \tilde{W}f_\theta(x_i)\| - \frac{1}{n}\sum_{i=1}^n \|\varphi(x_i)\|$$

$$= \frac{1}{n}\sum_{i=1}^n \|\tilde{W}f_\theta(\bar{x}_i)\| - \frac{1}{n}\sum_{i=1}^n \|\tilde{W}(f_\theta(\bar{x}_i) - f_\theta(x_i))\| - \frac{1}{n}\sum_{i=1}^n \|\varphi(x_i)\|$$

This implies that
$$\frac{1}{n}\sum_{i=1}^{n}\|\tilde{W}f_\theta(\bar{x}_i)\| \le L_S(w) + \frac{1}{n}\sum_{i=1}^{n}\|\tilde{W}(f_\theta(\bar{x}_i) - f_\theta(x_i))\| + \frac{1}{n}\sum_{i=1}^{n}\|\varphi(x_i)\|.$$

Furthermore, since $y = W^* f_\theta(x) + \varphi(x)$, by writing $\bar{y}_i = W^* f_\theta(\bar{x}_i) + \varphi(\bar{x}_i)$ (where $\bar{y}_i = y_i$ since $\bar{x}_i \sim \mathcal{D}_{y_i}$ for $i = 1, \ldots, n$),

$$\frac{1}{n}\sum_{i=1}^{n}\|\tilde{W}f_\theta(\bar{x}_i)\| = \frac{1}{n}\sum_{i=1}^{n}\|Wf_\theta(\bar{x}_i) - W^* f_\theta(\bar{x}_i)\|$$

$$= \frac{1}{n}\sum_{i=1}^{n}\|Wf_\theta(\bar{x}_i) - \bar{y}_i + \varphi(\bar{x}_i)\|$$

$$\ge \frac{1}{n}\sum_{i=1}^{n}\|Wf_\theta(\bar{x}_i) - \bar{y}_i\| - \frac{1}{n}\sum_{i=1}^{n}\|\varphi(\bar{x}_i)\|$$

Combining these, we have that

$$\frac{1}{n}\sum_{i=1}^{n}\|Wf_\theta(\bar{x}_i) - \bar{y}_i\| \le L_S(w) + \frac{1}{n}\sum_{i=1}^{n}\|\tilde{W}(f_\theta(\bar{x}_i) - f_\theta(x_i))\| \tag{32}$$

$$+ \frac{1}{n}\sum_{i=1}^{n}\|\varphi(x_i)\| + \frac{1}{n}\sum_{i=1}^{n}\|\varphi(\bar{x}_i)\|.$$

To bound the left-hand side of equation 32, we now analyze the following random variable:

$$\mathbb{E}_{X,Y}[\|W_S f_\theta(X) - Y\|] - \frac{1}{n}\sum_{i=1}^{n}\|W_S f_\theta(\bar{x}_i) - \bar{y}_i\|, \tag{33}$$

where $\bar{y}_i = y_i$ since $\bar{x}_i \sim \mathcal{D}_{y_i}$ for $i = 1, \ldots, n$. Importantly, this means that as $W_S$ depends on $y_i$, $W_S$ depends on $\bar{y}_i$. Thus, the collection of random variables $\|W_S f_\theta(\bar{x}_1) - \bar{y}_1\|, \ldots, \|W_S f_\theta(n_n) - \bar{y}_n\|$ is *not* independent. Accordingly, we cannot apply standard concentration inequality to bound equation 33. A standard approach in learning theory is to first bound equation 33 by $\mathbb{E}_{x,y}\|W_S f_\theta(x) - y\| - \frac{1}{n}\sum_{i=1}^{n}\|W_S f_\theta(\bar{x}_i) - \bar{y}_i\| \le \sup_{W \in \mathcal{W}} \mathbb{E}_{x,y}\|Wf_\theta(x) - y\| - \frac{1}{n}\sum_{i=1}^{n}\|Wf_\theta(\bar{x}_i) - \bar{y}_i\|$ for some hypothesis space $\mathcal{W}$ (that is independent of $S$) and realize that the right-hand side now contains the collection of independent random variables $\|Wf_\theta(\bar{x}_1) - \bar{y}_1\|, \ldots, \|Wf_\theta(n_n) - \bar{y}_n\|$, for which we can utilize standard concentration inequalities. This reasoning leads to the Rademacher complexity of the hypothesis space $\mathcal{W}$. However, the complexity of the hypothesis space $\mathcal{W}$ can be very large, resulting into a loose bound. In this proof, we show that we can avoid the dependency on hypothesis space $\mathcal{W}$ by using a very different approach with conditional expectations to take care the dependent random variables $\|W_S f_\theta(\bar{x}_1) - \bar{y}_1\|, \ldots, \|W_S f_\theta(n_n) - \bar{y}_n\|$. Intuitively, we utilize the fact that for these dependent random variables, there are a structure of conditional independence, conditioned on each $y \in \mathcal{Y}$.

We first write the expected loss as the sum of the conditional expected loss:

$$\mathbb{E}_{X,Y}[\|W_S f_\theta(X) - Y\|] = \sum_{y \in \mathcal{Y}} \mathbb{E}_{X,Y}[\|W_S f_\theta(X) - Y\| \mid Y = y]\mathbb{P}(Y = y)$$

$$= \sum_{y \in \mathcal{Y}} \mathbb{E}_{X_y}[\|W_S f_\theta(X_y) - y\|]\mathbb{P}(Y = y),$$

where $X_y$ is the random variable for the conditional with $Y = y$. Using this, we decompose equation 33 into two terms:

$$\mathbb{E}_{X,Y}[\|W_S f_\theta(X) - Y\|] - \frac{1}{n}\sum_{i=1}^{n}\|W_S f_\theta(\bar{x}_i) - \bar{y}_i\| \tag{34}$$

$$= \left(\sum_{y \in \mathcal{Y}} \mathbb{E}_{X_y}[\|W_S f_\theta(X_y) - y\|]\frac{|\mathcal{I}_y|}{n} - \frac{1}{n}\sum_{i=1}^{n}\|W_S f_\theta(\bar{x}_i) - \bar{y}_i\|\right)$$

$$+ \sum_{y \in \mathcal{Y}} \mathbb{E}_{X_y}[\|W_S f_\theta(X_y) - y\|]\left(\mathbb{P}(Y = y) - \frac{|\mathcal{I}_y|}{n}\right),$$

where
$$\mathcal{I}_y = \{i \in [n] : y_i = y\}.$$
The first term in the right-hand side of equation 34 is further simplified by using
$$\frac{1}{n} \sum_{i=1}^{n} \|W_S f_\theta(\bar{x}_i) - \bar{y}_i\| = \frac{1}{n} \sum_{y \in \mathcal{Y}} \sum_{i \in \mathcal{I}_y} \|W_S f_\theta(\bar{x}_i) - y\|,$$

as

$$\sum_{y \in \mathcal{Y}} \mathbb{E}_{X_y}[\|W_S f_\theta(X_y) - y\|] \frac{|\mathcal{I}_y|}{n} - \frac{1}{n} \sum_{i=1}^{n} \|W_S f_\theta(\bar{x}_i) - \bar{y}_i\|$$

$$= \frac{1}{n} \sum_{y \in \tilde{\mathcal{Y}}} |\mathcal{I}_y| \left( \mathbb{E}_{X_y}[\|W_S f_\theta(X_y) - y\|] - \frac{1}{|\mathcal{I}_y|} \sum_{i \in \mathcal{I}_y} \|W_S f_\theta(\bar{x}_i) - y\| \right),$$

where $\tilde{\mathcal{Y}} = \{y \in \mathcal{Y} : |\mathcal{I}_y| \neq 0\}$. Substituting these into equation equation 34 yields

$$\mathbb{E}_{X,Y}[\|W_S f_\theta(X) - Y\|] - \frac{1}{n} \sum_{i=1}^{n} \|W_S f_\theta(\bar{x}_i) - \bar{y}_i\| \tag{35}$$

$$= \frac{1}{n} \sum_{y \in \tilde{\mathcal{Y}}} |\mathcal{I}_y| \left( \mathbb{E}_{X_y}[\|W_S f_\theta(X_y) - y\|] - \frac{1}{|\mathcal{I}_y|} \sum_{i \in \mathcal{I}_y} \|W_S f_\theta(\bar{x}_i) - y\| \right)$$

$$+ \sum_{y \in \mathcal{Y}} \mathbb{E}_{X_y}[\|W_S f_\theta(X_y) - y\|] \left( \mathbb{P}(Y = y) - \frac{|\mathcal{I}_y|}{n} \right)$$

Importantly, while $\|W_S f_\theta(\bar{x}_1) - \bar{y}_1\|, \ldots, \|W_S f_\theta(\bar{x}_n) - \bar{y}_n\|$ on the right-hand side of equation 35 are dependent random variables, $\|W_S f_\theta(\bar{x}_1) - y\|, \ldots, \|W_S f_\theta(\bar{x}_n) - y\|$ are independent random variables since $W_S$ and $\bar{x}_i$ are independent and $y$ is fixed here. Thus, by using Hoeffding's inequality (Lemma 1), and taking union bounds over $y \in \tilde{\mathcal{Y}}$, we have that with probability at least $1 - \delta$, the following holds for all $y \in \tilde{\mathcal{Y}}$:

$$\mathbb{E}_{X_y}[\|W_S f_\theta(X_y) - y\|] - \frac{1}{|\mathcal{I}_y|} \sum_{i \in \mathcal{I}_y} \|W_S f_\theta(\bar{x}_i) - y\| \leq \kappa_S \sqrt{\frac{\ln(|\tilde{\mathcal{Y}}|/\delta)}{2|\mathcal{I}_y|}}.$$

This implies that with probability at least $1 - \delta$,

$$\frac{1}{n} \sum_{y \in \tilde{\mathcal{Y}}} |\mathcal{I}_y| \left( \mathbb{E}_{X_y}[\|W_S f_\theta(X_y) - y\|] - \frac{1}{|\mathcal{I}_y|} \sum_{i \in \mathcal{I}_y} \|W_S f_\theta(\bar{x}_i) - y\| \right)$$

$$\leq \frac{\kappa_S}{n} \sum_{y \in \tilde{\mathcal{Y}}} |\mathcal{I}_y| \sqrt{\frac{\ln(|\tilde{\mathcal{Y}}|/\delta)}{2|\mathcal{I}_y|}}$$

$$= \kappa_S \left( \sum_{y \in \tilde{\mathcal{Y}}} \sqrt{\frac{|\mathcal{I}_y|}{n}} \right) \sqrt{\frac{\ln(|\tilde{\mathcal{Y}}|/\delta)}{2n}}.$$

Substituting this bound into equation 35, we have that with probability at least $1 - \delta$,

$$\mathbb{E}_{X,Y}[\|W_S f_\theta(X) - Y\|] - \frac{1}{n} \sum_{i=1}^{n} \|W_S f_\theta(\bar{x}_i) - \bar{y}_i\| \tag{36}$$

$$\leq \kappa_S \left( \sum_{y \in \tilde{\mathcal{Y}}} \sqrt{\hat{p}(y)} \right) \sqrt{\frac{\ln(|\tilde{\mathcal{Y}}|/\delta)}{2n}} + \sum_{y \in \mathcal{Y}} \mathbb{E}_{X_y}[\|W_S f_\theta(X_y) - y\|] \left( \mathbb{P}(Y = y) - \frac{|\mathcal{I}_y|}{n} \right)$$

where

$$\hat{p}(y) = \frac{|\mathcal{I}_y|}{n}.$$

Moreover, for the second term on the right-hand side of equation 36, by using Lemma 1 of (Kawaguchi et al., 2022), we have that with probability at least $1 - \delta$,

$$\sum_{y \in \mathcal{Y}} \mathbb{E}_{X_y}[\|W_S f_\theta(X_y) - y\|] \left( \mathbb{P}(Y = y) - \frac{|\mathcal{I}_y|}{n} \right)$$

$$\leq \left( \sum_{y \in \mathcal{Y}} \sqrt{p(y)} \mathbb{E}_{X_y}[\|W_S f_\theta(X_y) - y\|] \right) \sqrt{\frac{2 \ln(|\mathcal{Y}|/\delta)}{2n}}$$

$$\leq \kappa_S \left( \sum_{y \in \mathcal{Y}} \sqrt{p(y)} \right) \sqrt{\frac{2 \ln(|\mathcal{Y}|/\delta)}{2n}}$$

where $p(y) = \mathbb{P}(Y = y)$. Substituting this bound into equation 36 with the union bound, we have that with probability at least $1 - \delta$,

$$\mathbb{E}_{X,Y}[\|W_S f_\theta(X) - Y\|] - \frac{1}{n} \sum_{i=1}^{n} \|W_S f_\theta(\bar{x}_i) - \bar{y}_i\| \tag{37}$$

$$\leq \kappa_S \left( \sum_{y \in \tilde{\mathcal{Y}}} \sqrt{\hat{p}(y)} \right) \sqrt{\frac{\ln(2|\tilde{\mathcal{Y}}|/\delta)}{2n}} + \kappa_S \left( \sum_{y \in \mathcal{Y}} \sqrt{p(y)} \right) \sqrt{\frac{2 \ln(2|\mathcal{Y}|/\delta)}{2n}}$$

$$\leq \left( \sum_{y \in \mathcal{Y}} \sqrt{\hat{p}(y)} \right) \kappa_S \sqrt{\frac{2 \ln(2|\mathcal{Y}|/\delta)}{2n}} + \left( \sum_{y \in \mathcal{Y}} \sqrt{p(y)} \right) \kappa_S \sqrt{\frac{2 \ln(2|\mathcal{Y}|/\delta)}{2n}}$$

$$\leq \kappa_S \sqrt{\frac{2 \ln(2|\mathcal{Y}|/\delta)}{2n}} \sum_{y \in \mathcal{Y}} \left( \sqrt{\hat{p}(y)} + \sqrt{p(y)} \right)$$

Combining equation 32 and equation 37 implies that with probability at least $1 - \delta$,

$$\mathbb{E}_{X,Y}[\|W_S f_\theta(X) - Y\|] \tag{38}$$

$$\leq \frac{1}{n} \sum_{i=1}^{n} \|W_S f_\theta(\bar{x}_i) - \bar{y}_i\| + \kappa_S \sqrt{\frac{2 \ln(2|\mathcal{Y}|/\delta)}{2n}} \sum_{y \in \mathcal{Y}} \left( \sqrt{\hat{p}(y)} + \sqrt{p(y)} \right)$$

$$\leq L_S(w_S) + \frac{1}{n} \sum_{i=1}^{n} \|\tilde{W}(f_\theta(\bar{x}_i) - f_\theta(x_i))\|$$

$$+ \frac{1}{n} \sum_{i=1}^{n} \|\varphi(x_i)\| + \frac{1}{n} \sum_{i=1}^{n} \|\varphi(\bar{x}_i)\| + \kappa_S \sqrt{\frac{2 \ln(2|\mathcal{Y}|/\delta)}{2n}} \sum_{y \in \mathcal{Y}} \left( \sqrt{\hat{p}(y)} + \sqrt{p(y)} \right).$$

We will now analyze the term $\frac{1}{n} \sum_{i=1}^{n} \|\varphi(x_i)\| + \frac{1}{n} \sum_{i=1}^{n} \|\varphi(\bar{x}_i)\|$ on the right-hand side of equation 38. Since $W^* = W_{\bar{S}}$,

$$\frac{1}{n} \sum_{i=1}^{n} \|\varphi(x_i)\| = \frac{1}{n} \sum_{i=1}^{n} \|g^*(x_i) - W_{\bar{S}} f_\theta(x_i)\|.$$

By using Hoeffding's inequality (Lemma 1), we have that for any $\delta > 0$, with probability at least $1 - \delta$,

$$\frac{1}{n} \sum_{i=1}^{n} \|\varphi(x_i)\| \leq \frac{1}{n} \sum_{i=1}^{n} \|g^*(x_i) - W_{\bar{S}} f_\theta(x_i)\| \leq \mathbb{E}_{x^+}[\|g^*(x^+) - W_{\bar{S}} f_\theta(x^+)\|] + \kappa_{\bar{S}} \sqrt{\frac{\ln(1/\delta)}{2n}}.$$

Moreover, by using (Mohri et al., 2012, Theorem 3.1) with the loss function $x^+ \mapsto \|g^*(x^+) - Wf(x^+)\|$ (i.e., Lemma 2), we have that for any $\delta > 0$, with probability at least $1 - \delta$,

$$\mathbb{E}_{x^+}[\|g^*(x^+) - W_{\bar{S}} f_\theta(x^+)\|] \leq \frac{1}{m} \sum_{i=1}^m \|g^*(x_i^+) - W_{\bar{S}} f_\theta(x_i^+)\| + \frac{2\tilde{\mathcal{R}}_m(\mathcal{W} \circ \mathcal{F})}{\sqrt{m}} + \kappa \sqrt{\frac{\ln(1/\delta)}{2m}}$$
(39)

where $\tilde{\mathcal{R}}_m(\mathcal{W} \circ \mathcal{F}) = \frac{1}{\sqrt{m}} \mathbb{E}_{\bar{S},\xi}[\sup_{W \in \mathcal{W}, f \in \mathcal{F}} \sum_{i=1}^m \xi_i \|g^*(x_i^+) - Wf(x_i^+)\|]$ is the normalized Rademacher complexity of the set $\{x^+ \mapsto \|g^*(x^+) - Wf(x^+)\| : W \in \mathcal{W}, f \in \mathcal{F}\}$ (it is normalized such that $\tilde{\mathcal{R}}_m(\mathcal{F}) = O(1)$ as $m \to \infty$ for typical choices of $\mathcal{F}$), and $\xi_1, \ldots, \xi_m$ are independent uniform random variables taking values in $\{-1, 1\}$. Takinng union bounds, we have that for any $\delta > 0$, with probability at least $1 - \delta$,

$$\frac{1}{n} \sum_{i=1}^n \|\varphi(x_i)\| \leq \frac{1}{m} \sum_{i=1}^m \|g^*(x_i^+) - W_{\bar{S}} f_\theta(x_i^+)\| + \frac{2\tilde{\mathcal{R}}_m(\mathcal{W} \circ \mathcal{F})}{\sqrt{m}} + \kappa \sqrt{\frac{\ln(2/\delta)}{2m}} + \kappa_{\bar{S}} \sqrt{\frac{\ln(2/\delta)}{2n}}$$

Similarly, for any $\delta > 0$, with probability at least $1 - \delta$,

$$\frac{1}{n} \sum_{i=1}^n \|\varphi(\bar{x}_i)\| \leq \frac{1}{m} \sum_{i=1}^m \|g^*(x_i^+) - W_{\bar{S}} f_\theta(x_i^+)\| + \frac{2\tilde{\mathcal{R}}_m(\mathcal{W} \circ \mathcal{F})}{\sqrt{m}} + \kappa \sqrt{\frac{\ln(2/\delta)}{2m}} + \kappa_{\bar{S}} \sqrt{\frac{\ln(2/\delta)}{2n}}.$$

Thus, by taking union bounds, we have that for any $\delta > 0$, with probability at least $1 - \delta$,

$$\frac{1}{n} \sum_{i=1}^n \|\varphi(x_i)\| + \frac{1}{n} \sum_{i=1}^n \|\varphi(\bar{x}_i)\| \tag{40}$$

$$\leq \frac{2}{m} \sum_{i=1}^m \|g^*(x_i^+) - W_{\bar{S}} f_\theta(x_i^+)\| + \frac{4\mathcal{R}_m(\mathcal{W} \circ \mathcal{F})}{\sqrt{m}} + 2\kappa \sqrt{\frac{\ln(4/\delta)}{2m}} + 2\kappa_{\bar{S}} \sqrt{\frac{\ln(4/\delta)}{2n}}$$

To analyze the first term on the right-hand side of equation 40, recall that

$$W_{\bar{S}} = \underset{W'}{\text{minimize}} \|W'\|_F \text{ s.t. } W' \in \arg\min_W \frac{1}{m} \sum_{i=1}^m \|Wf_\theta(x_i^+) - g^*(x_i^+)\|^2. \tag{41}$$

Here, since $Wf_\theta(x_i^+) \in \mathbb{R}^r$, we have that

$$Wf_\theta(x_i^+) = \text{vec}[Wf_\theta(x_i^+)] = [f_\theta(x_i^+)^\top \otimes I_r] \text{vec}[W] \in \mathbb{R}^r,$$

where $I_r \in \mathbb{R}^{r \times r}$ is the identity matrix, and $[f_\theta(x_i^+)^\top \otimes I_r] \in \mathbb{R}^{r \times dr}$ is the Kronecker product of the two matrices, and $\text{vec}[W] \in \mathbb{R}^{dr}$ is the vectorization of the matrix $W \in \mathbb{R}^{r \times d}$. Thus, by defining $A_i = [f_\theta(x_i^+)^\top \otimes I_r] \in \mathbb{R}^{r \times dr}$ and using the notation of $w = \text{vec}[W]$ and its inverse $W = \text{vec}^{-1}[w]$ (i.e., the inverse of the vectorization from $\mathbb{R}^{r \times d}$ to $\mathbb{R}^{dr}$ with a fixed ordering), we can rewrite equation 41 by

$$W_{\bar{S}} = \text{vec}^{-1}[w_{\bar{S}}] \quad \text{where} \quad w_{\bar{S}} = \underset{w'}{\text{minimize}} \|w'\|_F \text{ s.t. } w' \in \arg\min_w \sum_{i=1}^m \|g_i - A_i w\|^2,$$

with $g_i = g^*(x_i^+) \in \mathbb{R}^r$. Since the function $w \mapsto \sum_{i=1}^m \|g_i - A_i w\|^2$ is convex, a necessary and sufficient condition of the minimizer of this function is obtained by

$$0 = \nabla_w \sum_{i=1}^m \|g_i - A_i w\|^2 = 2 \sum_{i=1}^m A_i^\top (g_i - A_i w) \in \mathbb{R}^{dr}$$

This implies that

$$\sum_{i=1}^m A_i^\top A_i w = \sum_{i=1}^m A_i^\top g_i.$$

In other words,

$$A^\top A w = A^\top g \quad \text{where } A = \begin{bmatrix} A_1 \\ A_2 \\ \vdots \\ A_m \end{bmatrix} \in \mathbb{R}^{mr \times dr} \text{ and } g = \begin{bmatrix} g_1 \\ g_2 \\ \vdots \\ g_m \end{bmatrix} \in \mathbb{R}^{mr}$$

Thus,

$$w' \in \arg\min_w \sum_{i=1}^m \|g_i - A_i w\|^2 = \{(A^\top A)^\dagger A^\top g + v : v \in \text{Null}(A)\}$$

where $(A^\top A)^\dagger$ is the Moore–Penrose inverse of the matrix $A^\top A$ and $\text{Null}(A)$ is the null space of the matrix $A$. Thus, the minimum norm solution is obtained by

$$\text{vec}[W_{\bar{S}}] = w_{\bar{S}} = (A^\top A)^\dagger A^\top g.$$

Thus, by using this $W_{\bar{S}}$, we have that

$$\frac{1}{m} \sum_{i=1}^m \|g^*(x_i^+) - W_{\bar{S}} f_\theta(x_i^+)\| = \frac{1}{m} \sum_{i=1}^m \sqrt{\sum_{k=1}^r ((g_i - A_i w_{\bar{S}})_k)^2}$$

$$\leq \sqrt{\frac{1}{m} \sum_{i=1}^m \sum_{k=1}^r ((g_i - A_i w_{\bar{S}})_k)^2}$$

$$= \frac{1}{\sqrt{m}} \sqrt{\sum_{i=1}^m \sum_{k=1}^r ((g_i - A_i w_{\bar{S}})_k)^2}$$

$$= \frac{1}{\sqrt{m}} \|g - A w_{\bar{S}}\|_2$$

$$= \frac{1}{\sqrt{m}} \|g - A(A^\top A)^\dagger A^\top g\|_2 = \frac{1}{\sqrt{m}} \|(I - A(A^\top A)^\dagger A^\top)g\|_2$$

where the inequality follows from the Jensen's inequality and the concavity of the square root function. Thus, we have that

$$\frac{1}{n} \sum_{i=1}^n \|\varphi(x_i)\| + \frac{1}{n} \sum_{i=1}^n \|\varphi(\bar{x}_i)\| \tag{42}$$

$$\leq \frac{2}{\sqrt{m}} \|(I - A(A^\top A)^\dagger A^\top)g\|_2 + \frac{4\mathcal{R}_m(\mathcal{W} \circ \mathcal{F})}{\sqrt{m}} + 2\kappa \sqrt{\frac{\ln(4/\delta)}{2m}} + 2\kappa_{\bar{S}} \sqrt{\frac{\ln(4/\delta)}{2n}}$$

By combining equation 38 and equation 42 with union bound, we have that

$$\mathbb{E}_{X,Y}[\|W_S f_\theta(X) - Y\|] \tag{43}$$

$$\leq L_S(w_S) + \frac{1}{n} \sum_{i=1}^n \|\tilde{W}(f_\theta(\bar{x}_i) - f_\theta(x_i))\| + \frac{2}{\sqrt{m}} \|\mathbf{P}_A g\|_2$$

$$+ \frac{4\mathcal{R}_m(\mathcal{W} \circ \mathcal{F})}{\sqrt{m}} + 2\kappa \sqrt{\frac{\ln(8/\delta)}{2m}} + 2\kappa_{\bar{S}} \sqrt{\frac{\ln(8/\delta)}{2n}}$$

$$+ \kappa_S \sqrt{\frac{2\ln(4|\mathcal{Y}|/\delta)}{2n}} \sum_{y \in \mathcal{Y}} \left( \sqrt{\hat{p}(y)} + \sqrt{p(y)} \right).$$

where $\tilde{W} = W_S - W^*$ and $\mathbf{P}_A = I - A(A^\top A)^\dagger A^\top$.

We will now analyze the second term on the right-hand side of equation 43:

$$\frac{1}{n} \sum_{i=1}^n \|\tilde{W}(f_\theta(\bar{x}_i) - f_\theta(x_i))\| \leq \|\tilde{W}\|_2 \left( \frac{1}{n} \sum_{i=1}^n \|f_\theta(\bar{x}_i) - f_\theta(x_i)\| \right), \tag{44}$$

where $\|\tilde{W}\|_2$ is the spectral norm of $\tilde{W}$. Since $\bar{x}_i$ shares the same label with $x_i$ as $\bar{x}_i \sim \mathcal{D}_{y_i}$ (and $x_i \sim \mathcal{D}_{y_i}$), and because $f_\theta$ is trained with the unlabeled data $\bar{S}$, using Hoeffding's inequality (Lemma 1) implies that with probability at least $1 - \delta$,

$$\frac{1}{n} \sum_{i=1}^n \|f_\theta(\bar{x}_i) - f_\theta(x_i)\| \le \mathbb{E}_{y \sim \rho} \mathbb{E}_{\bar{x}, x \sim \mathcal{D}_y^2} [\|f_\theta(\bar{x}) - f_\theta(x)\|] + \tau_{\bar{S}} \sqrt{\frac{\ln(1/\delta)}{2n}}. \tag{45}$$

Moreover, by using (Mohri et al., 2012, Theorem 3.1) with the loss function $(x, \bar{x}) \mapsto \|f_\theta(\bar{x}) - f_\theta(x)\|$ (i.e., Lemma 2), we have that with probability at least $1 - \delta$,

$$\mathbb{E}_{y \sim \rho} \mathbb{E}_{\bar{x}, x \sim \mathcal{D}_y^2} [\|f_\theta(\bar{x}) - f_\theta(x)\|] \le \frac{1}{m} \sum_{i=1}^m \|f_\theta(x_i^+) - f_\theta(x_i^{++})\| + \frac{2\tilde{\mathcal{R}}_m(\mathcal{F})}{\sqrt{m}} + \tau \sqrt{\frac{\ln(1/\delta)}{2m}} \tag{46}$$

where $\tilde{\mathcal{R}}_m(\mathcal{F}) = \frac{1}{\sqrt{m}} \mathbb{E}_{\bar{S}, \xi}[\sup_{f \in \mathcal{F}} \sum_{i=1}^m \xi_i \|f(x_i^+) - f(x_i^{++})\|]$ is the normalized Rademacher complexity of the set $\{(x^+, x^{++}) \mapsto \|f(x^+) - f(x^{++})\| : f \in \mathcal{F}\}$ (it is normalized such that $\tilde{\mathcal{R}}_m(\mathcal{F}) = O(1)$ as $m \to \infty$ for typical choices of $\mathcal{F}$), and $\xi_1, \dots, \xi_m$ are independent uniform random variables taking values in $\{-1, 1\}$. Thus, taking union bound, we have that for any $\delta > 0$, with probability at least $1 - \delta$,

$$\frac{1}{n} \sum_{i=1}^n \|\tilde{W}(f_\theta(\bar{x}_i) - f_\theta(x_i))\| \tag{47}$$

$$\le \|\tilde{W}\|_2 \left( \frac{1}{m} \sum_{i=1}^m \|f_\theta(x_i^+) - f_\theta(x_i^{++})\| + \frac{2\tilde{\mathcal{R}}_m(\mathcal{F})}{\sqrt{m}} + \tau \sqrt{\frac{\ln(2/\delta)}{2m}} + +\tau_{\bar{S}} \sqrt{\frac{\ln(2/\delta)}{2n}} \right).$$

By combining equation 43 and equation 47 using the union bound, we have that with probability at least $1 - \delta$,

$$\mathbb{E}_{X,Y}[\|W_S f_\theta(X) - Y\|] \tag{48}$$

$$\le L_S(w_S) + \|\tilde{W}\|_2 \left( \frac{1}{m} \sum_{i=1}^m \|f_\theta(x_i^+) - f_\theta(x_i^{++})\| + \frac{2\tilde{\mathcal{R}}_m(\mathcal{F})}{\sqrt{m}} + \tau \sqrt{\frac{\ln(4/\delta)}{2m}} + \tau_{\bar{S}} \sqrt{\frac{\ln(4/\delta)}{2n}} \right)$$

$$+ \frac{2}{\sqrt{m}} \|\mathbf{P}_A g\|_2 + \frac{4\mathcal{R}_m(\mathcal{W} \circ \mathcal{F})}{\sqrt{m}} + 2\kappa \sqrt{\frac{\ln(16/\delta)}{2m}} + 2\kappa_{\bar{S}} \sqrt{\frac{\ln(16/\delta)}{2n}}$$

$$+ \kappa_S \sqrt{\frac{2\ln(8|\mathcal{Y}|/\delta)}{2n}} \sum_{y \in \mathcal{Y}} \left( \sqrt{\hat{p}(y)} + \sqrt{p(y)} \right)$$

$$= L_S(w_S) + \|\tilde{W}\|_2 \left( \frac{1}{m} \sum_{i=1}^m \|f_\theta(x_i^+) - f_\theta(x_i^{++})\| \right) + \frac{2}{\sqrt{m}} \|\mathbf{P}_A g\|_2 + Q_{m,n}$$

where

$$Q_{m,n} = \|\tilde{W}\|_2 \left( \frac{2\tilde{\mathcal{R}}_m(\mathcal{F})}{\sqrt{m}} + \tau \sqrt{\frac{\ln(3/\delta)}{2m}} + \tau_{\bar{S}} \sqrt{\frac{\ln(3/\delta)}{2n}} \right)$$

$$+ \kappa_S \sqrt{\frac{2\ln(6|\mathcal{Y}|/\delta)}{2n}} \sum_{y \in \mathcal{Y}} \left( \sqrt{\hat{p}(y)} + \sqrt{p(y)} \right)$$

$$+ \frac{4\mathcal{R}_m(\mathcal{W} \circ \mathcal{F})}{\sqrt{m}} + 2\kappa \sqrt{\frac{\ln(4/\delta)}{2m}} + 2\kappa_{\bar{S}} \sqrt{\frac{\ln(4/\delta)}{2n}}.$$

Define $Z_{\bar{S}} = [f(x_1^+), \dots, f(x_m^+)] \in \mathbb{R}^{d \times m}$. Then, we have $A = [Z_{\bar{S}}^\top \otimes I_r]$. Thus,

$$\mathbf{P}_A = I - [Z_{\bar{S}}^\top \otimes I_r][Z_{\bar{S}} Z_{\bar{S}}^\top \otimes I_r]^\dagger [Z_{\bar{S}} \otimes I_r] = I - [Z_{\bar{S}}^\top (Z_{\bar{S}} Z_{\bar{S}}^\top)^\dagger Z_{\bar{S}} \otimes I_r] = [\mathbf{P}_{Z_{\bar{S}}} \otimes I_r]$$

where $\mathbf{P}_{Z_{\bar{S}}} = I_m - Z_{\bar{S}}^\top (Z_{\bar{S}} Z_{\bar{S}}^\top)^\dagger Z_{\bar{S}} \in \mathbb{R}^{m \times m}$. By defining $Y_{\bar{S}} = [g^*(x_1^+), \dots, g^*(x_m^+)]^\top \in \mathbb{R}^{m \times r}$, since $g = \text{vec}[Y_{\bar{S}}^\top]$,

$$\|\mathbf{P}_A g\|_2 = \|[\mathbf{P}_{Z_{\bar{S}}} \otimes I_r] \text{vec}[Y_{\bar{S}}^\top]\|_2 = \|\text{vec}[Y_{\bar{S}}^\top \mathbf{P}_{Z_{\bar{S}}}]\|_2 = \|\mathbf{P}_{Z_{\bar{S}}} Y_{\bar{S}}\|_F \tag{49}$$

On the other hand, recall that $W_S$ is the minimum norm solution as

$$W_S = \underset{W'}{\text{minimize}} \, \|W'\|_F \text{ s.t. } W' \in \arg\min_W \frac{1}{n} \sum_{i=1}^n \|W f_\theta(x_i) - y_i\|^2.$$

By solving this, we have

$$W_S = Y^\top Z_S{}^\top (Z_S Z_S{}^\top)^\dagger,$$

where $Z_S = [f(x_1), \ldots, f(x_n)] \in \mathbb{R}^{d \times n}$ and $Y_S = [y_1, \ldots, y_n]^\top \in \mathbb{R}^{n \times r}$. Then,

$$
\begin{aligned}
L_S(w_S) = \frac{1}{n} \sum_{i=1}^n \|W_S f_\theta(x_i) - y_i\| &= \frac{1}{n} \sum_{i=1}^n \sqrt{\sum_{k=1}^r ((W_S f_\theta(x_i) - y_i)_k)^2} \\
&\leq \sqrt{\frac{1}{n} \sum_{i=1}^n \sum_{k=1}^r ((W_S f_\theta(x_i) - y_i)_k)^2} \\
&= \frac{1}{\sqrt{n}} \|W_S Z_S - Y^\top\|_F \\
&= \frac{1}{\sqrt{n}} \|Y^\top (Z_S{}^\top (Z_S Z_S{}^\top)^\dagger Z_S - I)\|_F \\
&= \frac{1}{\sqrt{n}} \|(I - Z_S{}^\top (Z_S Z_S{}^\top)^\dagger Z_S) Y\|_F
\end{aligned}
$$

Thus,

$$L_S(w_S) = \frac{1}{\sqrt{n}} \|\mathbf{P}_{Z_S} Y\|_F \tag{50}$$

where $\mathbf{P}_{Z_S} = I - Z_S{}^\top (Z_S Z_S{}^\top)^\dagger Z_S$.

By combining equation 48–equation 50 and using $1 \leq \sqrt{2}$, we have that with probability at least $1 - \delta$,

$$\mathbb{E}_{X,Y}[\|W_S f_\theta(X) - Y\|] \leq c I_{\bar{S}}(f_\theta) + \frac{2}{\sqrt{m}} \|\mathbf{P}_{Z_{\bar{S}}} Y_{\bar{S}}\|_F + \frac{1}{\sqrt{n}} \|\mathbf{P}_{Z_S} Y_S\|_F + Q_{m,n}, \tag{51}$$

where

$$
\begin{aligned}
Q_{m,n} = c &\left( \frac{2\tilde{\mathcal{R}}_m(\mathcal{F})}{\sqrt{m}} + \tau \sqrt{\frac{\ln(3/\delta)}{2m}} + \tau_{\bar{S}} \sqrt{\frac{\ln(3/\delta)}{2n}} \right) \\
&+ \kappa_S \sqrt{\frac{2 \ln(6|\mathcal{Y}|/\delta)}{2n}} \sum_{y \in \mathcal{Y}} \left( \sqrt{\hat{p}(y)} + \sqrt{p(y)} \right) \\
&+ \frac{4\mathcal{R}_m(\mathcal{W} \circ \mathcal{F})}{\sqrt{m}} + 2\kappa \sqrt{\frac{\ln(4/\delta)}{2m}} + 2\kappa_{\bar{S}} \sqrt{\frac{\ln(4/\delta)}{2n}}.
\end{aligned}
$$

$\square$

## H    KNOWN LEMMAS

We use the following well-known theorems as lemmas in our proof. We put these below for the completeness. These are classical results and *not* our results.

**Lemma 1.** (Hoeffding's inequality) *Let $X_1, \ldots, X_n$ be independent random variables such that $a \leq X_i \leq b$ almost surely. Consider the average of these random variables, $S_n = \frac{1}{n}(X_1 + \cdots + X_n)$. Then, for all $t > 0$,*

$$\mathbb{P}_S \left( \mathbb{E}[S_n] - S_n \geq (b-a) \sqrt{\frac{\ln(1/\delta)}{2n}} \right) \leq \delta,$$

*and*

$$\mathbb{P}_S \left( S_n - \mathrm{E}\left[S_n\right] \geq (b-a)\sqrt{\frac{\ln(1/\delta)}{2n}} \right) \leq \delta.$$

*Proof.* By using Hoeffding's inequality, we have that for all $t > 0$,

$$\mathbb{P}_S \left( \mathrm{E}\left[S_n\right] - S_n \geq t \right) \leq \exp\left( -\frac{2nt^2}{(b-a)^2} \right),$$

and

$$\mathbb{P}_S \left( S_n - \mathrm{E}\left[S_n\right] \geq t \right) \leq \exp\left( -\frac{2nt^2}{(b-a)^2} \right),$$

Setting $\delta = \exp\left( -\frac{2nt^2}{(b-a)^2} \right)$ and solving for $t > 0$,

$$1/\delta = \exp\left( \frac{2nt^2}{(b-a)^2} \right)$$

$$\Longrightarrow \ln(1/\delta) = \frac{2nt^2}{(b-a)^2}$$

$$\Longrightarrow \frac{(b-a)^2 \ln(1/\delta)}{2n} = t^2$$

$$\Longrightarrow t = (b-a)\sqrt{\frac{\ln(1/\delta)}{2n}}$$

$\square$

It has been shown that generalization bounds can be obtained via Rademacher complexity (Bartlett & Mendelson, 2002; Mohri et al., 2012; Shalev-Shwartz & Ben-David, 2014). The following is a trivial modification of (Mohri et al., 2012, Theorem 3.1) for a one-sided bound on the nonnegative general loss functions:

**Lemma 2.** *Let $\mathcal{G}$ be a set of functions with the codomain $[0, M]$. Then, for any $\delta > 0$, with probability at least $1 - \delta$ over an i.i.d. draw of $m$ samples $S = (q_i)_{i=1}^m$, the following holds for all $\psi \in \mathcal{G}$:*

$$\mathbb{E}_q[\psi(q)] \leq \frac{1}{m}\sum_{i=1}^m \psi(q_i) + 2\mathcal{R}_m(\mathcal{G}) + M\sqrt{\frac{\ln(1/\delta)}{2m}}, \tag{52}$$

*where $\mathcal{R}_m(\mathcal{G}) := \mathbb{E}_{S,\xi}[\sup_{\psi \in \mathcal{G}} \frac{1}{m}\sum_{i=1}^m \xi_i \psi(q_i)]$ and $\xi_1, \ldots, \xi_m$ are independent uniform random variables taking values in $\{-1, 1\}$.*

*Proof.* Let $S = (q_i)_{i=1}^m$ and $S' = (q_i')_{i=1}^m$. Define

$$\varphi(S) = \sup_{\psi \in \mathcal{G}} \mathbb{E}_{x,y}[\psi(q)] - \frac{1}{m}\sum_{i=1}^m \psi(q_i). \tag{53}$$

To apply McDiarmid's inequality to $\varphi(S)$, we compute an upper bound on $|\varphi(S) - \varphi(S')|$ where $S$ and $S'$ be two test datasets differing by exactly one point of an arbitrary index $i_0$; i.e., $S_i = S_i'$ for all $i \neq i_0$ and $S_{i_0} \neq S_{i_0}'$. Then,

$$\varphi(S') - \varphi(S) \leq \sup_{\psi \in \mathcal{G}} \frac{\psi(q_{i_0}) - \psi(q_{i_0}')}{m} \leq \frac{M}{m}. \tag{54}$$

Similarly, $\varphi(S) - \varphi(S') \leq \frac{M}{m}$. Thus, by McDiarmid's inequality, for any $\delta > 0$, with probability at least $1 - \delta$,

$$\varphi(S) \leq \mathbb{E}_S[\varphi(S)] + M\sqrt{\frac{\ln(1/\delta)}{2m}}. \tag{55}$$

Moreover,

$$\mathbb{E}_S[\varphi(S)] \tag{56}$$

$$= \mathbb{E}_S\left[\sup_{\psi\in\mathcal{G}} \mathbb{E}_{S'}\left[\frac{1}{m}\sum_{i=1}^{m}\psi(q_i')\right] - \frac{1}{m}\sum_{i=1}^{m}\psi(q_i)\right] \tag{57}$$

$$\leq \mathbb{E}_{S,S'}\left[\sup_{\psi\in\mathcal{G}} \frac{1}{m}\sum_{i=1}^{m}(\psi(q_i') - \psi(q_i))\right] \tag{58}$$

$$\leq \mathbb{E}_{\xi,S,S'}\left[\sup_{\psi\in\mathcal{G}} \frac{1}{m}\sum_{i=1}^{m}\xi_i(\psi(q_i') - \psi(q_i))\right] \tag{59}$$

$$\leq 2\mathbb{E}_{\xi,S}\left[\sup_{\psi\in\mathcal{G}} \frac{1}{m}\sum_{i=1}^{m}\xi_i\psi(q_i)\right] = 2\mathcal{R}_m(\mathcal{G}) \tag{60}$$

where the fist line follows the definitions of each term, the second line uses the Jensen's inequality and the convexity of the supremum, and the third line follows that for each $\xi_i \in \{-1, +1\}$, the distribution of each term $\xi_i(\ell(f(x_i'), y_i') - \ell(f(x_i), y_i))$ is the distribution of $(\ell(f(x_i'), y_i') - \ell(f(x_i), y_i))$ since $S$ and $S'$ are drawn iid with the same distribution. The forth line uses the subadditivity of supremum. $\square$

## I   SIMCLR

In contrastive learning, different augmented views of the same image are attracted (positive pairs), while different augmented views are repelled (negative pairs). MoCo (He et al., 2020) and SimCLR (Chen et al., 2020) are recent examples of self-supervised visual representation learning that reduce the gap between self-supervised and fully-supervised learning. SimCLR applies randomized augmentations to an image to create two different views, $x$ and $y$, and encodes both of them with a shared encoder, producing representations $r_x$ and $r_y$. Both rx and $r_y$ are $l2$-normalized. The SimCLR version of the InfoNCE objective is:

$$\mathbb{E}_{x,y}\left[-\log\left(\frac{e^{\frac{1}{\eta}r_y^T r_x}}{\sum_{k=1}^{K} e^{\frac{1}{\eta}r_{y_k}^T r_x}}\right)\right]$$

where $\eta$ is a temperature term and $K$ is the number of views in a minibatch

