# OpenReview forum: "What Do We Maximize in Self-Supervised Learning And Why Does Generalization Emerge?"
_ICLR.cc/2023/Conference — Submitted to ICLR 2023_

### Official Review · Reviewer_DWKP · 2022-10-22

**Confidence:** 4
**Correctness:** 3
**Technical Novelty And Significance:** 3
**Empirical Novelty And Significance:** 3
**Recommendation:** 6

**Clarity, Quality, Novelty And Reproducibility:**


The presented results are novel and interesting, but the assumptions are quite restrictive. The writing can be improved for better clarity. The results are likely reproducible.

**Strength And Weaknesses:**

**Strength**

The main strength of the paper is its justification of VICReg and the generalization bound provided.

**Weakness**

* The mixture of Gaussian assumption particularly considering the Gaussian components "do not overlap" is quite rough and lacks rigour. Any two Gaussians necessarily overlap.  When the overlapped region of two Gaussians has a small probability (under either one of the two Gaussians), which is presumably what the paper tries to assume by "do not overlap", the modes of the two Gaussians will need to be sufficiently apart.  But in many cases, for example when the data distribution is nearly uniform on its support, to model the data distribution using a mixture of Gaussians, a large number of significantly overlapping Gaussian would be needed, which violates the assumption of the paper.

* The presentation of paper seems sloppy, awkward and sometimes lacks clarity. I am listing a few here.
1. In Equation (3), the notation of raising the Gaussian density to the power of "$T=n$" is strange.
2. In beginning of section 4, when you say "when our input noise is small", what do you mean by "input noise"? Please also justify this statement, ie, why when the "input noise is small" you can reduce the density $p_{z|x^*}$ to a single Gaussian?
3. In section 6.1, the notation ${\cal T}$ appears multiple times, which is supposed to mean transpose?




**Summary Of The Paper:**

Under certain mixture of Gaussian assumption of the data distribution, the paper provides an information-theoretic justification of the VICReg method for self-supervised learning. Additionally, a generalization bound for the down-stream tasks is established.

**Summary Of The Review:**

The paper presents interesting and novel results, with some limitation,  that justify VICReg. A generalization bound for the downstream tasks is also given.  Clarity can be improved.

---

> ### Author Response · Authors · 2022-11-17
> **Response to reviewer DWKP**
>
> Thank you for your time and the great feedback. We note that in addition to this response post, we have also made a separate general post with several clarifications and new results inspired by your comments, including new experiments about the performance of new objective functions and the validity of assumptions. In addition, we have made a few changes to section 4 to clarify and extend the experimental section with our new results. We respond to each of your points below:
>
> **Overlapping of the Gaussian components assumption.**
>
> - __Rigorously__. As you pointed out, Gaussians have infinite support, so they must overlap with each other. However, the effective support is the domain of the distribution with probability greater than epsilon, Namely {x∈RD:p(x)>\eps}. According to our assumption, the Gaussian components are separated, i.e., their shared effective support becomes negligible (Huber et al., 2008 discussed a similar assumption).   We clarify this point in the revised manuscript.
>
> - __The validity of our assumptions.__ In response to your comment, we have added experiments to verify our assumption. The general response provides a detailed description and results. We want to emphasize that in these experiments, we found that the effective supports of our high-dimensional datasets were not overlapping, and our assumption is realistic even for current SSL datasets. We agree, however, that this assumption may be too restrictive in some specific scenarios. We have revised the manuscript and added the information and figures in response to your comments.
>
> **The presentation of the paper.** Thank you for pointing out these formatting mistakes. In our updated draft, we have corrected each of these errors. Additionally, we re-edited section 4 of the paper to make it more understandable and clear.
>
> **What do we mean by "input noise"?**
>
> Since the DN is linear within a small region w around x, then as long as the effective support of the Gaussian centered at x is contained within w, then p(z|x) is effectively a single Gaussian (since it is a Gaussian transformed by an affine transformation, i.e., another Gaussian with different parameters). So the exact condition for "the input noise is small" is that the effective support of the Gaussian centered at x is contained within the region w of the DN's input space partition.
>
> **References**
>
> [Huber et al., 2008]  On Entropy Approximation for Gaussian Mixture Random Vectors
>
>
> Thank you very much for your review. We have taken great care in responding to your concerns. We hope you will consider raising your score based on our response. Please let us know if you have additional questions we can address.

---

> > ### Author Response · Authors · 2022-12-07
> > **Response to Reviewer DWKP - Further engagement**
> >
> > Dear reviewer DWKP,
> >
> > Thank you again for your thoughtful review.
> >
> > Does our response address your questions? We would appreciate the opportunity to engage further if needed.

---

> > > ### Comment · Reviewer_DWKP · 2022-12-13
> > > **Thank you**
> > >
> > > Thank you for the response.
> > >
> > > I still have some reservation on the argument of "effective support" of Gaussian densities, particularly since small probability of the Gaussian tails can get magnified by the nonlinear network and may not be ignored at the network output. Nonetheless, I am maintaining my score. Looking at other reviews, I think my rating won't be the main obstacle for the acceptance of this paper.

---

### Official Review · Reviewer_KD44 · 2022-10-24

**Confidence:** 3
**Correctness:** 2
**Technical Novelty And Significance:** 4
**Empirical Novelty And Significance:** Not applicable
**Recommendation:** 3

**Clarity, Quality, Novelty And Reproducibility:**

* **Clarity**: The paper is clear in some parts but it misses some citations justifying some claims, important information such as when does a theorem hold, or how some equations are obtained. Also, the intent take-away of some parts of the text (e.g. Section 5) are not clear to me.

* **Quality**: The quality of the paper varies. While the proof of Theorem 2 is beautiful albeit with some assumptions that need to be included to make it work, the first part of the text is a little subpar, with the example of using an upper bound of the entropy in a lower bound to justify an objective.

* **Novelty**: The paper is definitely novel. Considering the input as a mixture of Gaussians, continuous piecewise affine (CPA) mappings, and both Theorem 2 and its proof is new and very interesting.

* **Reproducibility**: \
*Theory*: I reproduced all the proofs except of the questions that I placed the authors in the weaknesses.

  *Experiments*: There is no code to reproduce the experiments. Hence, I did not have time to replicate them. They should be "easy" to replicate due to the small size of the datasets and simplicity of the experiment. Nonetheless, it would be helpful to include this resource, both for the goodness of fit tests and the GMMs experiments.

**Strength And Weaknesses:**

**Strengths**

The paper includes a couple of nice ideas, for instance:

* Considering the input as a mixture of Gaussians allows them to analyze properties of the output under certain assumptions.
* Under the assumption that SSL with a stop-gradient operation is an Expectation-Maximization problem, one can try to avoid collapse explicitly.

The proof of the generalization bound for SSL algorithms is beautifully constructed and the resulting inequality is interesting.

* They go around having to deal with the large Rademacher complexity $\tilde{\mathcal{R}}(\mathcal{W})$ in a clever manner.
* They manage to include the decreasing term $c = \lVert W_S - W_{\bar{S}} \rVert$ in some of the terms of their inequality, where $W_S$ and $W_{\bar{S}}$ are the minimum norm solutions of the empirical risk on the labeled $S$ and unlabeled $\bar{S}$ datasets.

**Weaknesses**

* Sometimes the paper makes some claims without a citation to prior work:

    *  In the second paragraph of the introduction, it is claimed that some objective functions in the literature contradict each other, as well as many implicit assumptions. It would be good for the community to provide some examples of these cases.

    * In the first paragraph of Section 3.2., they claim that under the manifold hypothesis, any point can be seen as a Gaussian random variable with a low-rank covariance matrix in the direction of the manifold tangent space of the data. It would be nice to have some citation of such a hypothesis or some justification as of why this is true.

* There are some mis-representations of the literature.

    * The background section seems to suggest that no previous work has considered the input the source of randomness of deep neural networks. This is actually not the case. For instance, some works on contrastive learning have considered the network as deterministic and only the inputs as random variables, e.g. *[Zimmermann et al. 2021; Wang and Isola 2020]*.

* Sometimes the paper is either not rigorous or forgets to add important information to the reader.

    * In Section 3.2., what is the effective support of a random variable?. Also, in a standard dataset, are the supports of Gaussians centered at the images densities with non-overlapping supports?

    * Equation (4) is wrong since $p(x)$ is a density and $\mathcal{N}({x}; {x}^*[n({x})], \Sigma_{{x}*[n({x})]}) / N$ does not integrate to 1. I believe that you wanted to write that without dividing by $N$.

    * Section 3.3. it is mentioned that an analytical form of the per-region affine mappings is given in section 2. I could not find that. What did you mean?

    * Theorem 3 is stated as an absolute statement instead of an approximate statement. The assumption for the theorem is that $p({x}|{x}_{n({x})}^*)$ is approximately a Gaussian. Furthermore, in the proof it is also assumed that $\int_\omega p({x}|{x}^*[n({x})]) dx \approx 1$. Then, the linearity of the mapping with respect to $p$ is also an approximation.

    * In the first equation in Section 4, what are $\mu_n$ and $\Sigma_n$? I assume they are $\mu_n = A_{\omega(x_n)} x_n$ and $\Sigma_n = A_{\omega(x_n)}^T \Sigma_{x_n} A_{\omega(x_n)}$.

    * How is (7) obtained? Since the conditional decoder is the Gaussian $\mathcal{N}(\mu_n, \Sigma_r + \Sigma_n)$, the corresponding conditional entropy would be $$\frac{d}{2} \log(2 \pi e) + \frac{1}{2} \mathbb{E}[ \log |\Sigma_r + \Sigma_n|].$$  It is not clear to me how (7) is obtained from here.

    * In Section 4, it is said that $H(Z)$ and $H(Z|X')$ cannot increase if the other does not. It would be nice to explain why this happens, since it does not seem obvious to me. In fact it seems it could be false. While $Z|X'=x'$ is a Gaussian with covariance $\Sigma_r + \Sigma_n$, the marginal $Z$ is a mixture of Gaussians. One could keep the matrices $A$ fixed so that the covariance $\Sigma_n$ is the same and thus also the entropy $H(Z|X')$, and then modify the biases ${b}$ arbitrarily so the components of $Z$ are as far apart as possible, hence increasing the variance and thus the entropy $H(Z)$ arbitrarily.

    * Also in Section 4, it is mentioned that a simpler solution is to approximate the entire mixture as a single Gaussian by only capturing the first two moments and as this provides an upper bound of the entropy $H(Z)$. Why is this a good approximation? And why would this be a sensitive choice for the objective?

        1. As per your distribution model, the distribution of $Z$ is a mixture of Gaussians where the inputs were also Gaussians and well separated. Thus, this seems to indicate that $Z$ will be far from Gaussian or any unimodal-distributed random variable.

        2. Also, note that you want to maximize the mutual information $I(Z;X')$. For this reason you use the lower bound $I(Z;X') \geq H(Z) + \mathbb{E}[\log q(Z|X')]$. Then, if you use an upper bound on $H(Z)$ for your objective, you end up with an objective that is neither an upper nor a lower bound of $I(Z;X')$ and therefore maximizing it offers no guarantee that it maximizes the mutual information of interest.

  * In Section 4.3. and Figure 1 (left), the results suggest that the hypothesis that, for VICReg, the conditional output density is not Gaussian can be rejected with 85\% confidence for small noise and increases to 99\% confidence as the noise increases. This is fine but again, it means that the probability of a Type I error when rejecting the hypothesis that the data is not Gaussian is 15\%. \
  Also note that if the significance level were set to the standard $0.05$ the Gaussianity assumption would be rejected after the normalized noise is between 0.3 and 0.4. \
  I believe that these findings and caveats should be more clear in the text of that section.

  * Theorem 2 only holds for finite labels. In the notation of the main text and later in the proof it is mentioned that $y \in \mathbb{R}^r$. However, a crucial part of the proofs uses $p(y) = \mathbb{P}(Y=y)$ which only makes sense for discrete random variables. Then, it also includes the term $|\mathcal{Y}|$ which will result on a non-vacuous bound only if the amount of possible labels is finite. \
  Moreover, to employ *[Kawaguchi et al. 2022]* one needs that the random variable is multinomial and therefore finite. \
  Since one can always re-label the labels as they want this does not take away the nice proof, but I believe this fact should be made clear in the main text. The only constraint to do the re-labelling is that $\lVert y \rVert \leq \zeta$ for all $y \in \mathcal{Y}$.

  * Theorem 2 only holds for a classification (or regression of finite elements) with a linear layer and the $\ell_2$ norm as the loss. That is, for classification tasks where the loss is measured as $\lVert W f_\theta(X) - Y \rVert$. This is a very particular (and not common) scenario. I believe that this should be made clear as well in the main text.

  * There are some restrictions on the norms that need to be imposed for the Theorem to work.

    * Unless it is imposed that the class $\mathcal{F}$ has a finite norm range, i.e., $\mathcal{F} = \lbrace f_\theta: \mathcal{X} \to \mathcal{Z} \ | \ \lVert f_\theta(x) \rVert \leq a \textnormal{ for all } x \in \mathcal{X} \rbrace$ for some finite $a$, then $\tilde{\mathcal{R}}(\mathcal{F})$, $\tilde{\mathcal{R}}(\mathcal{W} \circ \mathcal{F})$, and $\kappa$ will tend to $\infty$ and the bounds will become vacuous.

    * Similarly, unless it is imposed that the class of matrices for the linear layer $\mathcal{W}$ is of finite norm, i.e., $\mathcal{W} = \lbrace w \in \mathbb{R}^{r \times K} \ | \ \lVert w \rVert \leq b \rbrace$ for some finite $b$, then both $\tilde{\mathcal{R}}(\mathcal{W} \circ \mathcal{F})$ and $\kappa$ will tend to $\infty$ and the bounds will become vacuous.

* It is not clear to me what is the insight to be gained in Section 5. I understand that *[Chen and He 2021]* made a connection between learning with a siamese network and a stop-gradient operation and expectation maximization ($k$-means in particular), but I do not see why noting how to avoid collapse for clustering with GMMs adds to the topic. The reason is that if you fix all covariances a priory to be the identity matrix, you need $\Sigma_{n} = A_{\omega(x_n)}^T \Sigma_{x} A_{\omega(x_n)} = I$, and therefore you cannot learn $A_{_{\omega(x_n)}}$. Am I missing something here? What is the purpose of this section?

**References**

*[Wang and Isola 2020]* Understanding contrastive representation learning through alignment and uniformity on the hypersphere. \
*[Zimmermann et al. 2021]* Contrastive learning inverts the data generating process. \
*[Chen and He 2021]* Exploring simple siamese representation learning. \
*[Kawaguchi et al. 2022]* On the theory of implicit deep learning: Global convergence with implicit layers.

**Summary Of The Paper:**

This paper intends to provide an understanding of self-supervised learning using information theory.

First, they lay down certain assumptions for deterministic networks, under which they conclude that deterministic networks' projections are distributed as a mixture of Gaussians.

Then, under the assumption from *[Federici et al. 2020]* that we want to maximize the mutual information $I(Z;X')$ and $I(Z';X)$ between the inputs $X$ and $X'$ and the projections $Z = f(X)$ and $Z' = f(X)$, they justify the objective of VICReg *[Bardes et al. 2021]*.

After that, they use the hypothesis from *[Chen and He 2021]* that says that a siamese network with a stop-gradient operation implements an Expectation-Maximization problem analogous to $k$-means. An iteration consists of a selection of $k$ clusters and an assignment of those. Under this assumption, they use the reduction of Gaussian mixture models (GMMs) clustering to $k$-means to show that one may avoid collapse enforcing sparse posteriors.

Finally, they conclude with a generalization bound for self-supervised methods using a linear layer for classification. This particularly exhibits that VICReg *[Bardes et al. 2021]* minimizes their lower bound on the generalization error.

**References**

*[Federici et al. 2020]* Learning robust
representations via multi-view information bottleneck. \
*[Chen and He 2021]* Exploring simple siamese representation learning.

**Summary Of The Review:**

This paper considers the hypothesis that the input data can be viewed as a mixture of Gaussians with low-rank covariances.

 Under that assumption, they attempt to justify the objective of VICReg.

   * If I am not mistaken, there is a flaw in their argument here and it is that their argument starts saying that they want to maximize $I(Z;X')$. They find a lower bound on it, and then they use an upper bound on one of the terms of said lower bound to optimize $I(Z;X')$. Hence, they are optimizing a quantity that is neither an upper nor a lower bound of their objective. As mentioned in the weaknesses, this may not even be a good estimate.

It has been hypothesized that learning using siamese networks with a stop-gradient operation is equivalent to an expectation-maximization algorithm (in particular $k$-means). Thus they make a suggestion on how to avoid collapse on clustering with a Gaussian mixture model (GMM).

  * Even though the idea is nice, it is unclear to me, though, how this can be directly linked back to self-supervised learning or learning with siamese networks with a stop-gradient operation.

Finally, they present a new bound on the generalization of SSL methods for classification tasks and a linear layer on top of the learned representations.

  * The proof of the bound is pretty and the result is interesting. However, it is not clearly stated the restrictions under which such theorem applies.

Overall, I believe that this paper has very interesting ideas and a beautiful proof of a nice theorem. However, it has some lack of rigor and some mistakes in certain places. Also, it requires some work to downplay some of the claims such as "different SSL models can be (re)discovered based on...", when only VICReg is re-discovered. All in all, I believe that the paper is not quite ready **yet**.

For this reason, at the current state, I must recommend rejection of the paper. Nonetheless, I encourage the authors to continue down this path since interesting results seem to await. Of course, if the issues discussed in the weaknesses section can be addressed satisfactorily during the rebuttal phase, I am open to increase the score, since a proper application of the ideas in this paper would be a good contribution.

**Minor comments and nitpicks that did not impact the score of the review**

* In Section 3.1, note that since $Z$ is a continuous random variable and assuming that it has a density the (differential) entropy goes to $- \infty$ not to $0$ as the Shannon entropy would.

* In Section 3.3. it is mentioned that the transformation goes from a space of dimension $D$ to a space of dimension $K$ where $K \geq D$. Usually the input space (images) is much larger than the space of the projections or representations, i.e. $D \gg K$.

* In Section 4, before (6) I believe you wanted to write $q(z|z') \sim \mathcal{N}(z',\Sigma_r)$.

* Throughout the text sometimes you write the vectors $x_n$ and $x_n^*$ in bold and sometimes without bold. It would be better to have a consistent notation.

* In the last paragraph of Section 2 the citation of *[Achille and Soatto 2018]* should be citet.

* Right before Section 4.1. it the citation of *[Moshksar and Khandani 2016]* should be with citep.

* If I am not mistaken you are sometimes writing $A \mathcal{T}$ to refer to a transpose and sometimes $A^T$. Please, could you use the same notation throughout? If you use the notation $A \mathcal{T}$, could you write it in some notation section as I do not recall this being standard.

* In the proof of Theorem 2 in the appendix, the notation $g^*$ is used without introduction (at least I did not find it). Could you include it?

* In certain point in the proof of Theorem 2 in the appendix (end of page 19), you started forgetting to include the tilde in the Rademacher complexity of $\mathcal{W} \circ \mathcal{F}$.

**References**

*[Achille and Soatto 2018]* Information dropout: Learning optimal representations
through noisy computation. \
*[Moshksar and Khandani 2016]* Arbitrarily tight bounds on differential entropy of Gaussian mixtures.

---

> ### Author Response · Authors · 2022-11-17
> **Response to reviewer KD44**
>
> Thank you for your time and valuable feedback. We want to point out that in addition to this response post, we have also made a separate general post, which contains several clarifications and new results based on your comments, including new experiments about the performance of new objective functions and the validity of assumptions. Your comments were highly valuable, and we are confident they have improved our updated manuscript. We respond to each of your points below:
>
> **Citation of prior work.**
>
> Based on your comments, we added several references to the updated manuscript. References were added for works that proposed different (and sometimes contradictory) information theory-based objectives. For example, although Kahana and Hoshen (2022) and Haoqing et al. (2022) suggest adding a term to maximize the information, others suggest minus the same term to minimize the information [Kuang-Huei et al., .2021).   Additionally, we added references [Fefferman et al., 2016, Liu et al., 2019]], which examine the manifold hypothesis both empirically and analytically, including developing algorithms for testing it and learning this manifold.
>
> **The literature regarded the input as the source of randomness in deep neural networks.**
>
>  We thank the reviewer for bringing these two relevant references to our attention. Despite Zimmermann et al. (2022) and Wang and Isola (2020) assuming a randomly generated input, they do not use any properties of the network's distribution itself. As opposed to this, our approach relies on the Gaussian properties of the network's output to derive our objective. A short discussion regarding these relevant references has been added to our manuscript.
>
> **What is the effective support of a random variable?**
>
> The effective support is the domain of the distribution with a probability greater than epsilon. According to our assumption, the Gaussian components are separated, i.e., their shared effective support becomes negligible (Huber et al., 2008 discussed a similar assumption).
>
> **The validity of our non-overlapping assumption.**
>
> Based on your comment, we have added experiments on seven datasets to verify our assumption. A detailed description and results are provided in the general response. In these experiments, we found that the effective supports of high-dimensional datasets did not overlap, and our assumption is valid for current SSL datasets. The manuscript has been revised, and the information and figures have been added in response to your comments.
>
> **Equation 4 normalization. ** We still believe that in order for it to integrate to 1, we need the 1/N term. Please let us know if you still think we need to correct something.
>
> **Missing the analytical form of the per-region affine mappings.**
>
> We thank you for this comment. In fact, we do not require the analytical knowledge of the per-region affine mapping, and its practical computation can be obtained by setting A to the Jacobian matrix of the network at the corresponding input x, and b to be defined as $f(x)-Ax$. This note has been added to the updated manuscript.
>
> **Theorem 3 is stated as an absolute statement instead of an approximate statement.**  We want to thank you for this comment and correct it in the revised manuscript.
>
>  **µ(x) and Σ(x).** These terms are as you wrote. We emphasized it in the text.
>
> **How is equation 7 obtained?**
>
> This equation is based on taking a lower bound on the log loss over the samples Z instead over the input X (which may be intractable). This term is still a gaussian but with a much simpler structure to decompose. The proof is based on decomposing Z' to  Z′=μ(x′)+L(x′)ε where ε∼N(0,1) and $L(x′)^T L(x′) = Σ(x′)$. We changed section 4 to clarify how we derived our objective based on your comment, and we hope it is much better now. A complete derivation is also included in appendix B. Thank you for this comment, which makes our manuscript much easier to read and understand.

---

> > ### Author Response · Authors · 2022-11-17
> > **Response to reviewer KD44 - Part 2**
> >
> > **Can H(Z) increase without changing   H(Z|X′)?**
> >
> > This is a great question. The relationship between the individual Gaussian components' entropies and the mixture's entropy is explained in detail in Huber et al. (2008). In sum, the mixture's entropy can be bounded by the Gaussian components' individual entropies. An upper bound for the entropy of a Gaussian mixture can be derived from the weighted sum of the entropies of the Gaussian components [Theorem 3 in  Huber et al., (2008)]. This upper bound is usually significantly closer to the true entropy than a single Gaussian matching the first two moments. When the Gaussian components are separated, the bound becomes exact [ Huber et al., 2008 ]. Your example showed a bound based on matching the first two moments of the Gaussian mixture. This bound is indeed independent of the entropy of individual Gaussian components and can be increased arbitrarily; however, it does not reflect the true entropy (or even the bound based upon the Gaussian component entropies). Please let us know if we have misunderstood your comment. It was helpful to receive your comments, which helped clarify this issue in the manuscript. In response to your comment, we have clarified the main text and added a detailed discussion of the entropy estimators and bounds to appendix C.
> >
> > **The approximation of the entropy by the first two moments as objective.**
> >
> > Thank you for this comment, which helps us to clarify a key component in our paper. Our information-theoretic analysis does not claim that VICReg is the most sensitive objective. Instead, we aim to expose the underlying assumptions and approximations of current self-supervised learning methods. For example, estimating the entropy of Z provides a good illustration of these approximations. A significant advantage of the bound used by VICReg to estimate the entropy (a single Gaussian matching the first two moments) is its simplicity and efficiency.   Despite this, previous studies have demonstrated that it has several disadvantages. First, as you mentioned, Z is a mixture of Gaussians with well-separated components, which means the Gaussian approximation is probably poor.
> > Moreover, this is an upper bound on the true entropy (and not even a tight one), which converges to an exact solution only for a single gaussian. As a result, maximizing an upper bound does not provide any guarantee regarding the true entropy, and in theory, this bound can be increased arbitrarily without affecting the true entropy. In practice, it may be possible to attain good results by maximizing a lower bound [Martinez et al., 2021, Nowozin et al., 2016], even if this may result in training instability. This is an example of the strength of our framework, which allows us directly compare and analyze whether our approximation methods are aligned with our assumptions. As we mentioned in the general comment, using other estimators of entropy yields better results, which is a promising area for future research. We have included this important discussion in section 4 and a detailed description of the different estimators and bounds in Appendix C, together with their advantages and disadvantages.
> >
> > **Comments regarding the significance test for the conditional output density.**  We agree with these comments and have incorporated them into the manuscript.
> >
> > **The cases where Theorem 2  holds.**  We agree with this point and emphasize it in the updated manuscript.
> >
> > **The restrictions on the norms for Theorem 2.** - We agree with your comment and added it to the main text. Thank you for these observations.
> >
> > **The purpose of section 5.**
> >
> > The purpose of section 5 is to relate another type of SSL method to our information optimization framework. We suggest analyzing the stop gradient SSL methods from the perspective of the dual optimization problem of EM, which was presented by Neal and Hinton (1998). A connection between these types of SSL methods and information was not presented by Chen and He (2021) and has several advantages. Integrating these SSL methods into our unified information-theoretic framework emphasizes the common issues associated with different SSL methods; how to prevent the collapse of the entropy of the marginal representation. Further, analyzing it as an information EM problem can provide us with several new observations and exciting research directions, particularly regarding ways to prevent this collapse. It is true that if all covariances are fixed a priori to be the identity matrix, you cannot learn Aω(x). However, as demonstrated in the manuscript, one way to prevent collapse is to enforce sparse posteriors, i.e., one-to-one mapping of the data points to the Gaussian's centers. Moreover, we also presented another method for preventing collapse - by varying the learning rates for the parameters and the input (between the two networks in the joint embedding SSL framework).

---

> > > ### Author Response · Authors · 2022-11-17
> > > **Response to reviewer KD44 - Part 3**
> > >
> > >
> > > **Experiments reproducibility.** Our code will be published soon.
> > >
> > > **Minor comments.** We fixed all the minor mistakes and typos based on your comments.
> > >
> > > **References**
> > >
> > > [Kahana and Hoshen, 2022] A Contrastive Objective for Learning Disentangled Representations \
> > > [Haoqing et al., 2022] Rethinking minimal sufficient representation in contrastive learning \
> > > [Kuang-Huei et al., .2021] Compressive visual representations \
> > > [Fefferman et al., 2016] Testing the manifold hypothesis \
> > > [Liu et al., 2019] Parametric Manifold Learning of Gaussian Mixture Models \
> > > [Huber et al., 2008]  On Entropy Approximation for Gaussian Mixture Random Vectors \
> > > [Martinez et al., 2021] Permute, Quantize, and Fine-tune: Efficient Compression of Neural Networks \
> > > [Nowozin et al., 2016] f-GAN: Training Generative Neural Samplers using Variational Divergence Minimization \
> > > [ Neal and Hinton, 1998] A View of the Em Algorithm that Justifies Incremental, Sparse, and other Variants \
> > >
> > > Thank you very much for your review. This review makes the paper much better and easy to follow. We have taken great care in responding to your concerns. We hope you will consider raising your score based on our response. Please let us know if you have additional questions we can address.

---

> ### Comment · Reviewer_KD44 · 2022-12-05
> **Answer to Responses**
>
> Thank you for your response and for taking the time to update the paper. Please, find below some more comments based on that.
>
> **Citation of prior work**
>
> It is not clear to me how these works contradict each other, sorry. I would need further clarification of such claims, especially in the paper, so it is clear to the reader.
>
> In the paper you write "Under the manifold hypothesis, any point can be seen as a Gaussian random variable with a low-rank covariance matrix in the direction of the manifold tangent space of the data (Fefferman et al., 2016)." However, the cited paper does not mention that. In fact, they describe the manifold hypothesis as "high dimensional data tend to lie near a low dimensional manifold", and put as an example data laying close to a 2d-torus. Also, I believe there is no [Liu et al, 2019] paper cited.
>
> **Equation (4) normalization**
>
> If $\mathcal{N}(x;\mu, \Sigma)$ is the density of a Gaussian distribution, it integrates to 1. Then, $p(x) = \mathcal{N}(x;\mu, \Sigma) / N$ integrates to $1/N$, so it is not a valid density. Correct me if I am wrong.
>
> **Missing the analytical form of the per-region affine mappings.**
>
> Can you explain why can we set $A_\omega$ to be the Jacobian at that point?
>
> **The approximation of the entropy by the first two moments as objective.**
>
> The paper does not claim that VICReg is the most sensible objective, but it says that it recovers it from first principles. There is no proof that VICReg is maximizing the mutual information, as you use a lower bound on the entropy instead of an upper bound.
>
> I still find that the idea of studying networks using the Gaussian mixture hypothesis on the samples and then maximizing the mutual information is valuable. But it is incorrect to say that VICReg is an instance of this.
>
> **The purpose of section 5**
>
> It is still not clear to me how there is a connection with information here. Could you elaborate?
>
> Also, what is the connection of sparse posterior in KNN or the learning rate approach with information?

---

> > ### Author Response · Authors · 2022-12-07
> > **Response to Reviewer KD44**
> >
> >  Thank you for your response. See our comments below:
> >
> > **Citation of prior work.**
> >
> > As a result of suggestions that our citations of prior work may be confusing, we have revised our wording in order to indicate that many of the works use different objective functions that are not clearly related and use different notations; Many of the previous works use different objective functions without a clear connection between them. Some of these works, such as [Hjelm et al. (2018), Henaff (2020), Dubois et al. (2021), Tian et al. (2020), and Tsai et al., 2020] suggest different objectives related to information, but the connection between these objectives is unclear. As we mentioned in the paper, the works of Kahana and Hoshen (2022) and Haoqing et al. (2022) suggest adding a term to maximize information, while Kuang-Huei et al. (2021) suggest minimizing the same term. We hope that this revised explanation provides a more precise understanding, and we will add it to the updated version. Thank you again for your feedback.
> >
> > **The manifold hypothesis.**
> >
> > This is exactly what we meant. When the author says "lie near a low dimensional manifold", this is equivalent to saying that the covariance matrix of the Gaussian centered at x has a low-rank covariance matrix, e.g., of the form epsilon $I+P^P$ where $P$ is a basis of the tangent space of the manifold at x. As a result, we believe that the verbatim of the reference and our sentence carry the same message. Thank you for noticing that the reference of  Liu et al. (2019) is missing. We will add it to the updated version of our manuscript.
> >
> > **Equation (4) normalization.**
> >
> > The normalization is necessary because when you integrate over x, you will integrate each N(x;μ, Σ) to 1, but you have N of them (one for each training point). The renormalization ensures that the overall integral of p(x) over the domain of x is equal to 1.
> >
> > **Missing the analytical form of the per-region affine mappings - why can we set Aω to be the Jacobian at that point?**
> >
> > Because the mapping is piecewise affine, i.e., for each region w, the output is obtained from Aωx+bω, and the Jacobian of the mapping is itself Aω. This comes directly from the fact that the DN input-output mapping is continuous piecewise affine.
> >
> > **The approximation of the entropy by the first two moments as objective.**
> >
> > In the paper, we are analyzing VICReg from an information theory perspective. We start with the theoretical goal of maximizing information and show the approximation and estimation along the way. Our framework allows us to address questions about the objective to optimize and how it aligns with our assumptions. The answer to the question of whether it is reasonable to maximize information in this way depends on the specific case. As we discussed in the paper, in theory, there are reasons to believe that optimizing an upper bound is better. However, in practice,  maximizing a lower bound on the information has produced good results in some cases [Martinez et al., 2021, Nowozin et al., 2016] even without restrictive assumptions. A lower bound maximization is one of many situations in which we may obtain suboptimal results. For example, optimizing a loose upper bound. It has been shown in the paper that using more "reasonable" estimators (both upper bounds and together) yields better results.
> >
> > **The purpose of section 5.**
> >
> > Although EM is usually thought of in terms of changing the parameters of the target function P, Neal and Hinton show how to view it as a dual optimization of the target distribution P and an auxiliary distribution Q. Moreover, they show that an EM iteration corresponds to maximizing the choice Q while holding P fixed and then maximizing P while holding Q fixed. As explained by Elidan and Friedman (2003), this is equivalent to maximizing the Lagrangian of the information. Viewing it as an information maximization problem emphasizes the common issues associated with different SSL methods and can provide new observations and exciting research directions. In the paper, we also showed how methods like the sparse posterior in KNN and the learning rate approach control the entropy and prevent it from collapsing.
> >
> > **References**
> >
> > [Hjelm et al, 2018] Learning deep representations by mutual information estimation and maximization \
> > [Henaff 2020] Data-efficient image recognition with contrastive predictive coding \
> > [Tian et al, 2020] Contrastive multiview coding \
> > [Tsai et al,  2020] Self-supervised learning from a multi-view perspective \
> > [Dubois et al, 2021] Lossy compression for lossless prediction \
> > [Elidan and Friedman, 2003] The Information Bottleneck EM Algorithm
> >
> >
> > Thank you again for your thoughtful review. Does our response address your questions? We would appreciate the opportunity to engage further and clarify any remaining concerns you may have.

---

### Official Review · Reviewer_48RM · 2022-10-25

**Confidence:** 2
**Clarity, Quality, Novelty And Reproducibility:** 1. Not sure where the p(z' | z) Gauss…
**Correctness:** 4
**Technical Novelty And Significance:** 4
**Empirical Novelty And Significance:** Not applicable
**Recommendation:** 5

**Strength And Weaknesses:**

Strengths
====
1. It is always valuable to cast heuristic objectives under a new and more formal perspective/framework, which this paper achieves.
2. Most of the discussion in the paper are fairly intuitive and easy to follow (at least until the derivation of VicReg).
3. The paper also presents a neat generalization bound for SSL that offers better guarantees than prior work (in terms of dependence on number of classes, and on the labeled dataset size). The bound depends on the three main terms of VicReg, thus providing a neat theoretical justification of VicReg. The proof of this result appears non-trivial (I've not checked it).

Weaknesses
====

My main concern is that I am not convinced of the significance of the results, however "neat" they seem to be. (Admittedly, this may also be due to the fact that I do not directly work in information theory.)

1. Although the information-theoretic view seems novel in that no one has pinned it down this way before, it is unclear to me what the significance is. When one examines the VICReg objective, it is fairly natural to believe that some kind of mutual information is being maximized: decreasing the off-diagonal terms of the covariance of Z while increasing the diagonal terms must naturally correspond to increasing the entropy H(Z); bringing Z and Z' closer together should correspond to reducing H(Z|Z'). The core connection in itself, I'm afraid, is not surprising.

2. Although the generalization bound makes a neat connection to the pieces of VicReg and is highly non-trivial (for which I greatly appreciate the authors' effort), it's unclear how significant of an insight this bound is.

3. The paper seems to claim that its information-theoretic formulate ``fully recovers the VICReg objective''. But is it right to say that the information-theoretic VICReg objective doesn't actually match the exact VICReg objective? e.g., maximizing $\log |\Sigma_Z|$ would correspond to maximizing the $\log$ of the diagonal variance terms, but in VICReg, this $\log$ wouldn't be there?

4. The paper makes a lot of *strong* (and novel) distributional assumptions to derive the existing objective. I understand that this is necessary to (a) resolve well-known information-theoretic quantities into well-known loss functions and (b) deal with a deterministic network. Given that (b) is novel, it'd be nice if (b) was also a technically significant contribution, which again is not clear to me.
- One question to the authors is, without these Gaussian assumptions, if we were forced to deal with a stochastic network, could one establish similar connections between VicReg and the mutual information maximization problem?




**Summary Of The Paper:**

1. This paper presents an information-theoretic view of SSL methods showing that current methods happen to maximize meaningful information-theoretic quantities.
2. The paper also presents a generalization bound on downstream tasks based on the above information-theoretical quantities.
3. Crucial to these results is the fact that the paper assumes a certain kind of stochasticity in the input, which allows information-theoretic results for a deterministic network (rather than a stochastic network as is typical).

**Summary Of The Review:**

The paper has neat results establishing the existing SSL objectives (VICReg & SimCLR) in theoretical foundations. However, I am not convinced of the significance of these results, at least to the ICLR community, as the takeaway in the paper is essentially "VicReg does make sense from a formal point of view". In an ideal world, I'd have hoped to see this paper (a) derive an insight that tells us something _new_ about how SSL works OR (b) suggested new algorithms. Or perhaps the paper simply hasn't laid out its own significance well enough or I'm missing something due to my lack of expertise in information theory.

---

> ### Author Response · Authors · 2022-11-17
> **Response to reviewer 48RM**
>
> Thank you for your time and feedback! It is greatly appreciated that you found our paper valuable and novel.  In addition to this response, we have published a separate general post. In this post,  we explain the unique contributions of our work, its significance, and new experiments of our proposed new objective functions that emerge from our analysis. Our updated manuscript has been greatly improved due to your helpful comments. \
> We respond to each of your points below:
>
>
> **The significance of the information-theoretic framework**
>
> Even though it is intuitively clear that we would like to maximize the information associated with the representations, structuring it rigorously has several goals; in addition to the general utility of a formal, unified framework for self-supervised methods, it also enables the identification, analysis, and comparison of the underlying assumptions and approximations of the various methods, thus enabling the development of better algorithms. The estimation of Z's entropy in our objective is a good example. As described in the manuscript, VICReg uses a single Gaussian matching for the first two moments. The main advantage of this method is its simplicity and efficiency.   However, previous research has demonstrated that it is a loose approximation that converges to the true entropy only for a single Gaussian [Huber et al., 2008]. Additionally, this approximation is an upper bound on the true entropy, which implies that we are optimizing an upper bound without a formal guarantee regarding optimizing the true entropy, resulting in instability during training.
>
> As described in the general comment, we have conducted new experiments demonstrating new methods for optimizing our information-theoretic objective based on different entropy estimators. Our two proposed estimators improve VICReg's original method, SimCLR and Barlow Twin on CIFAR-10. We would like to thank you for your comments, which helped us to improve our paper. Also, we added a discussion including various estimators and bounds, as well as their advantages and disadvantages, to Appendix C.
>
>
> **The significance of the generalization bound.**
>
> Please see our detailed comment above in the general response. Shortly, our bound, which is the first bound to address the VICReg objective,  provides a unique connection between our information-theoretic understanding of SSL and its generalization ability. Additionally, it enables us to better understand the superior generalization abilities of VICReg over supervised learning and SimCLR, as well as its unique properties.
>
> **Fully recover the VICReg objective.**
>
>  Yes, you are correct. There is no log on the covariance term of the VICreg objective. Therefore, it used an even looser approximation of Z's entropy than the one based on capturing the first two moments. There may be stability issues when optimizing log(x) when many xs are pushed toward zero. This is an example of how our framework can identify the approximation that current models are doing, as discussed above. We have added a discussion regarding entropy estimators and bounds to appendix C. Thanks for bringing this to our attention, and we have added the clarification to the revised manuscript.
>
> **The contribution of analyzing information in deterministic networks.**
>
> It has been demonstrated by Amjad et al. (2019) that in deep deterministic networks, the information is either infinite or piecewise constant, making gradient-based optimization methods impractical. Researchers have addressed these problems in various ways, including stochastic deep networks, which ensure that information is finite, and auxiliary DNN frameworks, which add additive noise to the model [Goldfeld et al., 2019, Elead et al., 2019]. In practice, however, most SSL methods are deterministic. To analyze and compare existing methods via information theory, we must first define what information is. We impose constraints on the data distribution, which helps us to be able to analyze deterministic networks directly from an information-theoretic perspective, which was previously not possible. We have revised the introduction in response to your comments.
>
> **ViCReg and stochastic networks.**
>
>  It is a very good question. As previously discussed, stochastic networks derive their randomness from the training procedure. As a result, we do not have to assume that samples are drawn from a specific distribution and can analyze information directly, assuming that the randomness comes from the network. Moreover, since the conditional output is Gaussian by definition in standard stochastic networks, we are not required to approximate this term as we did for the deterministic network. Training with stochastic networks, however, requires some modifications, including calculating the DKL and sampling. It would be interesting to compare the stochastic VICReg with the stochastic SimCLR [Lee et al., 2021].

---

> > ### Author Response · Authors · 2022-11-17
> > **Response to reviewer 48RM - Part 2**
> >
> > **The p(z' | z) Gaussian assumption.**
> >
> > All the regression tasks that use the l2 loss function implicitly assume that the decoder is a Gaussian around the output with variance 1. This assumption is unrelated to our other assumptions. In the supervised case, assuming a Gaussian observation model means $y=f(x) + \epsilon$ when epsilon is a gaussian with mean 0 and variance 1. For this observation model, the max likelihood estimator boils down to the l2 loss. For simplicity, and because $\Sigma_r$ is a constant, section 4 is now presenting the case where the covariance matrix  $\Sigma_r$ is equal to the identity matrix. As a result of your comment, we attempted to clarify this assumption better in the updated manuscript.
> >
> > **A more formal argument for the argument in section 4.1.** Thank you for your comment. Section 4 of our paper has been updated to make it more formal and easy to follow. Please let us know if this section is clear enough.
> >
> > **Presenting the VicReg and SimCLR objective.** We have included the VICReg objective in the background section, while we added the SimCLR objective to Appendix I (due to a lack of space).
> >
> > **Generalization bound assumptions.** The generalization bound does not require any assumptions made in the previous parts, such as the Gaussian assumption or input distribution assumption.
> >
> >
> > **References**
> >
> > [Saunshi et al., 2019], A theoretical analysis of contrastive unsupervised representation learning \
> > [Amjad et al., 2019] Learning representations for neural network-based classification using the information bottleneck principle \
> > [Elead et al., 2019] The effectiveness of layer-by-layer training using the information bottleneck principle \
> > [Lee et al, 2021] Compressive Visual Representations \
> > [Huber et al., 2008] On Entropy Approximation for Gaussian Mixture Random Vectors \
> > [Goldfeld et al., 2019] Estimating Information Flow in Deep Neural Networks
> >
> >
> > Thank you again for your comments on our paper. They have improved it. We hope you can consider raising your score in light of our response. We believe we are making many timely and significant contributions, outlined in the general response – and are happy to answer any further questions you might have.

---

> > > ### Author Response · Authors · 2022-12-07
> > > **Response to Reviewer 48RM - Further engagement**
> > >
> > > Dear reviewer 48RM,
> > >
> > > Thank you again for your thoughtful review.
> > >
> > > Does our response address your questions? We would appreciate the opportunity to engage further if needed.

---

> > > ### Comment · Reviewer_48RM · 2022-12-07
> > > **Thanks for the response**
> > >
> > > Hi authors,
> > >
> > > Apologies for the delay in acknowledging your responses.
> > >
> > > **I wish to maintain my score, but I think this paper is very promising if it could present its contributions clearly (which requires significant rewriting)**. Also, I personally felt that the responses didn't do justice in highlighting the key changes made to the paper and how it connects to our reviews.
> > >
> > > I've a few comments which I sincerely hope you may find helpful at least for future versions:
> > >
> > > - Rev KD44 had pointed out the issue with the final VicReg objective that is derived here from first principles is neither a lower bound nor an upper bound. I found this to be concerning, but it seemed to me that the response failed to address this head-on. It seems like the current objective is neither an upper bound nor a lower bound. I think both the paper and the responses could have been explicit about the fact that the overall term is neither a lower bound nor an upper bound. For future versions, I hope it's possible to explore a way to make this either a lower or an upper bound.
> > > - The updated paper has added **new preliminary experiments which are interesting** reportedly showing that replacing the entropy term with more sound approximations from first principles, provides improved results in a CIFAR10 setting. I think if the authors can make these experiments broader (try a few other datasets and architectures?), the paper would be significantly strengthened as it provides evidence for why your first principles analysis is valuable!
> > > - Having said that, the paper currently aims to do many things, which ideally should be a strength; but unfortunately, it fails to highlight them clearly which therefore makes it a weakness.
> > >     - One way to help this would be to enumerate the contributions more prominently and in a bit more detail in the introduction e.g., The three line summary at the end does not cover Sec 5. It does not summarize why the generalization bound is interesting compared to Saunshi et al.
> > >     - It might also be worth deciding which parts of the paper still need to be in the main paper. From my point of view, the first principles derivation, the corresponding empirical results, and the generalization bound together seem to be the most important contributions.
> > >     - When the paper says it unifies multiple SSL algorithms, it would be good to highlight which algorithms are unified and which sections they correspond to.  Is it only VicReg and Simclr?
> > >    - On a related note, the original version of the paper had a discussion that extended the analysis to SimClr, which is now missing. Is there a reason for this? Since the paper claims to be unifying various techniques, I'd love to see more than just VicReg analyzed. In fact, it would be interested if it's possible to derive novel objectives (different from VicReg variations) under varied assumptions.
> > >
> > >
> > > Other very minor stylistic suggestions for the future for a more effective response:
> > > - It may have helped to highlight the added discussions and sections in a different color. Similarly, it would have helped to bolden key phrases in the response (such as "added new experiments") so that important information stands out.
> > > - The new experiments involving the log term etc., could have been highlighted directly in the response to my question regarding ``Fully recover the VICReg objective.'' (besides in the longer common response).

---

> > > > ### Author Response · Authors · 2022-12-10
> > > > **Response to Reviewer 48RM**
> > > >
> > > >  Thank you for your response. See our comments below:
> > > >
> > > > **VICReg objective as an upper bound:** In this paper, we analyze VICReg from an information theory perspective. We start by showing that the theoretical goal of maximizing information can be approximated and estimated in our framework. This allows us to address questions about the objective function and how it aligns with our assumptions. The answer to the question of whether it is reasonable to maximize information in this way depends on the specific case. As discussed in the paper, there are theoretical reasons to believe optimizing an upper bound is better. However, in practice, maximizing a lower bound on the information has produced good results in some cases [Martinez et al., 2021, Nowozin et al., 2016] even without restrictive assumptions. A lower bound maximization is one of many situations where suboptimal results may be obtained. For example, optimizing a loose upper bound. Our experiments show that using more "reasonable" estimators (both upper bounds and together) yields better results.
> > > >
> > > >
> > > >
> > > > **Broader experiments:** Thank you for expressing interest in our new experiments. Our paper aims to provide a unified framework and theoretical insights. Of course, better experimental results on as many datasets as possible are always desirable. However, the empirical experiments in this paper serve only to support our claims and assumptions. We plan to apply our methods to more datasets, but this is outside the scope of this paper.
> > > >
> > > >
> > > > **Enumerate the contributions of the paper:**  Thank you for this suggestion. We agree that we could be more clear about the unique contributions of the paper, and we will add this to the introduction in the updated version.
> > > >
> > > > **Deciding which part of the paper should be in the main paper:** Thank you for this comment. We are a bit confused. Does it look like the only part that should be removed is section 5? The purpose of section 5 was to connect specific types of SSL models (the ones that use stop gradient) to our information maximization framework. Although EM is usually thought of in terms of changing the parameters of the target function P, Neal and Hinton show how to view it as a dual optimization of the target distribution P and an auxiliary distribution Q.
> > > > Moreover, they show that an EM iteration corresponds to maximizing the choice Q(T I Y) while holding P fixed and then maximizing P while holding Q(T I Y) fixed. As explained by Elidan and Friedman (2003), this is equivalent to maximizing the Lagrangian of the information. Viewing it as an information maximization problem emphasizes the common issues associated with different SSL methods and can provide new observations and exciting research directions.
> > > >
> > > >
> > > > **Unify multiple SSL algorithms:** Our framework unified every SSL algorithm that uses the InfoNCE algorithm (SimCLR, Ashman, et al., 2019), van den Oord et al. (2021)), the ones that use volume preserving methods such as VICReg and Barlow Twin (Zbontar et al., 2021)  (that is the same as VICReg but without the fixed variance term) and the stop gradients algorithms (BYOL (Gril et al., 2020), SimSiam,(Chen et al., 2021) and He et al. (2020))
> > > >
> > > > **Comparing to SimCLR:** Indeed, a paragraph compared SimCLR to VICReg and discussed their assumptions and differences. We removed it to lack of place in the main paper, but it might be worth bringing it back. Regarding the noble objectives, it would be an exciting direction to mix their assumptions of them and derive new objectives.
> > > >
> > > > Thank you also for your suggestions for the future response!
> > > >
> > > > **References**
> > > >
> > > > [van den Oord et al, 2021] Representation Learning with Contrastive Predictive Coding \
> > > > [achman et al, 2019]  Learning Representations by Maximizing Mutual Information Across Views \
> > > > [Chen et al, 2020] A Simple Framework for Contrastive Learning of Visual Representations \
> > > > [Zbontar et al, 2021] Barlow Twins: Self-Supervised Learning via Redundancy Reduction \
> > > > [Gril et al, 2020] Bootstrap your own latent: A new approach to self-supervised Learning \
> > > > [Chen et al, 2021] Exploring Simple Siamese Representation Learning \
> > > > [He et al, 2020] Momentum Contrast for Unsupervised Visual Representation Learning
> > > >
> > > >
> > > > Thank you again for your thoughtful review. Does our response address your questions? We would appreciate the opportunity to engage further and clarify any remaining concerns you may have.

---

### Official Review · Reviewer_3Mx6 · 2022-11-03

**Confidence:** 2
**Correctness:** 2
**Technical Novelty And Significance:** 3
**Empirical Novelty And Significance:** 2
**Recommendation:** 5

**Clarity, Quality, Novelty And Reproducibility:**

**Clarity:** The motivation of the paper is clear, but the details of the contribution need to be more clearly communicated.

**Novelty:** The contribution appears novel.

**Reproducibility:** I did not see any overt effort towards reproducibility, and the experimental setup seems to lack details.

**Strength And Weaknesses:**

### Strengths

1. **Clarity.** The paper is extremely well-written. I had no trouble understanding the ideas presented in the work and their relationship to prior work.

1. **Reasonable motivation.** The paper aims to derive SSL algorithms from an information-theoretic perspective, given that naive formulations thereof are ill-defined. This is reasonable and of interest to the community.

1. **Novelty.** Using a neural-networks-as-splines argument to derive an information-theoretic SSL optimization criterion appears novel.

### Weaknesses

1. **Assumptions are insufficiently related to practice.**
Broadly, the manuscript aims to explain SSL in practice (judging from the introduction).

    1. In Section 3.2, two assumptions are made regarding a model of the data distribution
    (non-overlapping effective support) and sufficient Gaussian modes.
    In practical settings (i.e., not in the limit of modes), how well does this represent data
    distributions of interest to SSL?

    1. The mapping $f$ is assumed *not* to be a dimensionality reduction operator (i.e., $f : D \to K$ with $K \ge D$).
    Is this not false of the neural networks used as standard in SSL?

1. **Unconvincing result.** The assumption of non-overlapping effective support in Theorem 1 appears to allow the output distribution of the neural network to depend quite simply as the result of a single affine spline (with parameters $A_\omega$, $b_\omega$).
This is in contrast to the more complicated expressions in Balestriero & Baraniuk (2018), who define splines per layer (e.g., their Eq. (6)).
Could you explain this discrepancy?
How could your setting "be extended to the general case," as is claimed in Section 3.2?

1. **No practical experiments.** Despite aiming to address SSL in practice, the paper does not attempt to optimize the bound on the MI objective derived in (7) nor empirically verify the generalization bound derived in Theorem 2.

### Minor

1. Please number all equations, etc. in the manuscript.

1. Around Eq. (2), $X$, $X'$, $Z$, $Z'$ are not explicitly defined, and it is not given which distribution to take the expectation with respect to (for completeness, though this is a standard expression)

1. The plot in Figure 1 looks stretched rather than scaled.

1. Typo (both "first" & "second"): "VICReg ... estimates the entropy of Z solely from the first second moment"

1. The "Theorem" counters in the appendix are off.

**Summary Of The Paper:**

The submission explores information-theoretically motivated objectives for self-supervised learning (SSL).
The submission derives a bound on an objective to maximize the mutual information between neural network inputs and outputs using assumptions about Gaussianity of the input distribution and a spline framing of neural network functions as in Montufar et al. (2014) and Balestriero & Baraniuk (2018).
A small experiment tests deviations from the normality of the distribution of outputs of a neural network, given inputs from a KDE-like expansion of the CIFAR-10 test dataset to validate the theory's assumptions.
A generalization bound and relationships to prior criteria for SSL optimization (spec. VICReg) are also derived.

**Summary Of The Review:**

The idea appears novel, but the practical significance of the results for SSL in general remains unclear.

---

> ### Author Response · Authors · 2022-11-17
> **Response to reviewer 3MX6**
>
> Thank you for your thoughtful and encouraging feedback! We appreciate that you found the paper to be extremely well-written and novel. Please note that in addition to this response post, we have also made a separate general post describing the unique contributions of this work, clarifying several points, and presenting new results based on your comments. Below are our responses to each of your points:
>
>
> **"The assumptions are not sufficiently related to practice"**
> - __The assumptions regarding the data model distribution.__ We agree that the validity of our assumptions is an important issue to consider. Based on your comment, we have added experiments to verify our assumptions. A detailed description and results are provided in the general response. In these experiments, we found that the effective supports of many high-dimensional datasets did not overlap, and our assumption is valid for many SSL datasets.  It is important to emphasize that in section 3.2, we only assume that any sample can be viewed as a Gaussian random variable centered at this point where their effective support does not overlap with nearby Gaussians substantially. Essentially, we are not assuming "sufficient Gaussian modes". In practice, the number of Gaussians equals the number of samples.  We want to thank you for this comment. Following your comments, we have revised the manuscript and added the information and the new figures.
>
> - __The assumption about the reduction operator.__ The validity of this assumption would depend on the application. For example, this mapping is not preserved in the case where D represents the ambient input space dimension. Our result, however, can be extended to include D as the intrinsic dimension of the data, where K>=D is a valid assumption for many cases. In this case, as long as the output dimension is equal to the intrinsic dimension of the data manifold, it is possible to define a one-to-one mapping so that the model's output around x remains a single Gaussian. Thank you for your important comment. We included a note in the updated manuscript.
>
> - __“Unconvincing results about the assumption of non-overlapping effective support which allows the output distribution of the neural network to depend quite simply as the result of a single affine spline”__. You are correct that our output distribution is quite simple. Because we assume localized support of the Gaussians centered at each data point, we obtain a simpler result which is exact for CPA networks, i.e., models with (leaky-)ReLU activation and the likes, and that is approximate for smooth models (from an approximation argument and first-order Taylor expansion). Because we are only interested in the per-sample Jacobian matrix and bias vector, we do not require explicitly obtaining the input space partition and other more involved quantities obtained in Balestriero & Baraniuk (2018) and for which a more involved derivation would be needed. Regarding extending our theory behind the non-overlapping Gaussian. In this case, the output of the network becomes a mixture of truncated Gaussian, which is more complicated to analyze, but still can be done similarly to the work in Balestriero et al., (2020).
>
> ***Practical experiments***
>
> - __Optimizing the information bound.__   Based on your comments, we have conducted new experiments demonstrating different methods for optimizing our information-theoretic objective. A detailed description and results are provided in the general response. Our proposed estimators improve VICReg's original methods and SimCLR and Barlow Twin on CIFAR-10. Thanks for your comments, which helped us to improve our paper.
> - __Calculating the generalization bound__ The biggest challenge for the generalization bound we derived in section 6 is calculating the  Rademacher complexity. Rademacher complexity measures how well functions in a function class fit random labels, which increases with class complexity. The Rademacher complexity depends on the functional class considered, so we must select a function class that captures only the real trained networks. Many attempts have been made to bound the Rademacher complexity for neural networks; however, these attempts were based on restrictive assumptions and were limited to oversimplified networks [Bartlett et al., 2017, Neyshabur et al., 2017]. Due to this, empirical evaluation of this bound is left for the future.
>
> ***Experiments reproducibility***.  Our code will be published soon.
>
> ***Minor comments.*** Thank you for pointing out these typos and editing errors. In our updated draft, we have corrected each of these errors.

---

> > ### Author Response · Authors · 2022-11-17
> > **Response to reviewer 3MX6 - Part 2**
> >
> >
> > ***References*** \
> > [Saunshi et al., 2019], A theoretical analysis of contrastive unsupervised representation learning\
> > [Balestriero et al., 2020] Analytical Probability Distributions and Exact Expectation-Maximization for Deep Generative Networks\
> > [Huber et al., 2008] On Entropy Approximation for Gaussian Mixture Random Vectors\
> > [Kolchinsky et al., 2017] Estimating Mixture Entropy with Pairwise Distances\
> > [Bartlett et al., 2017] Spectrally-normalized margin bounds for neural networks \
> > [Neyshabur et al., 2017] Exploring generalization in deep learning
> >
> >
> > Thank you again for your thoughtful and supportive review. Your comments have improved our paper. We made a significant effort to address your questions and would appreciate it if you would consider raising your score in light of our response. Also, please let us know if you have additional questions we can address.

---

> > > ### Author Response · Authors · 2022-12-07
> > > **Response to Reviewer 3MX6 - Further engagement**
> > >
> > > Dear reviewer 3MX6,
> > >
> > > Thank you again for your thoughtful review.
> > >
> > > Does our response address your questions? We would appreciate the opportunity to engage further if needed.

---

### Author Response · Authors · 2022-11-17
**General Response to Reviewers and AC**

We want to thank the reviewers for their thoughtful and supportive reviews. Their comments improved the manuscript and made it much better. We here provide a general response, addressed to all reviewers and ACs, as well as individual replies to address specific reviewer concerns as separate posts.

We want to emphasize the timeliness and importance of this work. First, an information-theoretic framework for SSL methods is presented. By shifting the required stochasticity to the input, we can analyze deterministic networks and, specifically, current SSL methods. Using our framework, we reveal and compare the fundamental underlying assumptions of the current SSL methods and suggest new ones based on the vast literature of information theory. We verify our assumptions using a variety of empirical evaluations. Based on our analysis, we also proposed new objectives that showed superior results compared to current methods. In addition, we wish to emphasize the novelty of our generalization bound for SSL, which is the first to address the VICReg objective, thus providing a theoretical understanding of  SSL and connecting our information-theoretic framework to the probabilistic guarantee of generalization ability. Moreover, our bound emphasizes the key components that make VICRreg more generalizable compared to supervised and other SSL methods.

Inspired by reviewer comments, we have now run several new experiments. These experiments are presented below, along with a discussion of several points raised by reviewers:

***Optimizing the information-theoretic objective with new methods***


Implementing our information-theoretic objective function in practice requires many "design choices". Based on the reviewer's suggestion, we conducted new experiments to empirically optimize our information-theoretic bound by combining the VICReg invariance term with different methods for entropy optimization. There has been extensive work on estimating the entropy of Gaussian mixtures without a clear solution. Our experiment used the Logarithm Determinant Entropy Estimator [Zhanghao and  Ding, 2021], which provides a tighter upper compared to the VICReg objective. However, this estimator is still an upper bound on entropy, which does not provide any guarantee. To address this problem, we also use a lower bound based on the pairwise distances of the individual Gaussians' means [Kolchinsky et al., 2017]. We run SSL training on CIFAR10 using ReNet18 and check the performance using linear evaluation, comparing our new estimators with both the original VICReg, SimCLR, and Barlow Twin. Our full results are presented in Table 1 in the appendix of our manuscript.
In summary, our proposed estimators outperform both the original VICReg and SimCLR and Barlow Twin. As we can see, by estimating the entropy with more accurate estimators, we can improve the results of VICReg. Moreover, the pairwise distance estimator, which is a lower bound, achieves the best results. This result aligns with the theory that we want to maximize a lower bound on true entropy. Based on our framework, we found that a wise selection of entropy estimators leads to better results. The updated manuscript contains our results and full technical details.

***The validity of our assumptions***

We agree that the validity of our assumptions is a very important issue. Therefore, we have added more experiments to verify our assumptions regarding the data model distribution. In these experiments,  we compute the pairwise ℓ2 distances between images for seven datasets: MNIST, CIFAR10, CIFAR100, Flowers102, Food101, and FGVAircaft. We found that even for raw pixels, the pairwise distances are far from zero, so you can use a small Gaussian around each point without overlapping. Consequently, the effective supports of these high-dimensional datasets are not overlapping, and our assumption is realistic even for popular SSL datasets.

We note that our manuscript contains another experiment that validates our assumptions. In this experiment, we check if the conditional output density reduces to a single Gaussian with decreasing input noise.  We showed that for small noise, we can reject the hypothesis that the conditional output density of the network is not Gaussian while increasing the noise causes the network's output to become less Gaussian.

---

> ### Author Response · Authors · 2022-11-17
> **General Response to Reviewers and AC - Part 2**
>
> ***Choices in implementation of the objective***
>
> Implementing our information-theoretic objective requires many "design choices" and approximations. In the paper, we showed that different methods employ different approaches to these design choices and approximations. The strength of our framework lies in its ability to formalize the different design choices, compare them, and analyze them within the broader context of information theory. In our analysis, we do not claim that one method is better than the other, but rather that our framework can directly compare and analyze whether our approximations along the why are aligned with our assumptions.   One example is estimating the entropy of  Z.  In VICReg, an approximation that is both loose and upper bound on the true entropy is used. As we presented above, as a result of our information-theoretic principles, we can use the broad range of previous works and use different estimators and bounds to improve the performance. The EM algorithm, presented in section 5, is another example. By analyzing the factors that cause this simple case to collapse, we can inspire ideas for better SSL training, such as different learning rates for each encoder to keep the entropy high enough.
>
> ***The significance of our generalization bound***
>
> In this paper, we propose a novel generalization bound for SSL, which is the first to address the VICReg objective, providing a theoretical foundation for SSL's information-theoretic understanding and its generalization ability. Furthermore, our bound emphasizes VICReg's advantages over supervised and other SSL methods. For example, in contrast to the supervised training's generalization bound, in which the hypothesis space complexity decreases as the labeled data size increases, our bound decreases as the self-supervised data size increases. The results demonstrate that, by using SSL, we can learn all the complexity of our tasks that were originally required by supervised examples with those from self-supervised examples (without labels) that usually significantly exceed the number of supervised examples. Furthermore, this bound emphasizes one of the most significant advantages of self-supervised learning - the importance of the information contained in the self-supervised examples. This fundamental dependency, which does not exist in SimCLR's previous generalization bound [Saunshi et al., 2019], enables us to understand better VICReg's superior generalization abilities over supervised learning and SimCLR. In addition,  our bound suggest new directions for improvement. For example, when we have partial information about the labels, we can use it to align the labels' covariance matrix and directly minimize a tighter bound.
>
>
> **References**
>
> [Zhanghao and  Ding, 2021] Understanding neural networks with logarithm determinant entropy estimator\
> [Kolchinsky et al., 2017] Estimating mixture entropy with pairwise distances

---

### Decision · Program_Chairs · 2023-01-20

**Decision:**

Reject

**Justification For Why Not Higher Score:**

The reviewers are mostly concerned with the significance of this work.

**Justification For Why Not Lower Score:**

N/A

**Metareview: Summary, Strengths And Weaknesses:**

This paper provides an information-theoretic framework for self-supervised learning (SSL), which unifies several existing SSL methods. It also derives a generalization bound for SSL methods, and provides generalization guarantees for the downstream supervised learning task.

Strengths:

+ The paper is well written
+ A generalization error bound for SSL is derived.

Weaknesses:
- Assumptions made in this paper are loosely connected with practice
- Certain parts of the paper lack rigorous argument.
- It is not clear how the information theoretic understanding can be used to design new SSL methods. So the significance of this work may be limited.
- No real convincing experimental support.

After the author's response and discussion, the reviewers are still not fully convinced regarding the significance of this work. The information-theoretic perspective of SSL is promising, but this idea needs a more thorough execution.